# Power-law adaptation in the presynaptic vesicle cycle

Fabian A. Mikulasch[1], Svilen V. Georgiev[2,3], Lucas Rudelt[1,4], Silvio O. Rizzoli [2,5,6] & Viola Priesemann [1,4,6] ✉

After synaptic transmission, fused synaptic vesicles are recycled, enabling the synapse to recover its capacity for renewed release. The recovery steps, which range from endocytosis to vesicle docking and priming, have been studied individually, but it is not clear what their impact on the overall dynamics of synaptic recycling is, and how they influence signal transmission. Here we model the dynamics of vesicle recycling and find that the multiple timescales of the recycling steps are reflected in synaptic recovery. This leads to multi-timescale synapse dynamics, which can be described by a simplified synaptic model with 'power-law' adaptation. Using cultured hippocampal neurons, we test this model experimentally, and show that the duration of synaptic exhaustion changes the effective synaptic recovery timescale, as predicted by the model. Finally, we show that this adaptation could implement a specific function in the hippocampus, namely enabling efficient communication between neurons through the temporal whitening of hippocampal spike trains.

One of the main bottlenecks of synaptic transmission is the recovery of synaptic vesicles after vesicle release, since it determines the availability of releasable vesicles. By now there is ample data on the underlying biochemical processes that proceed through several stages of vesicle recovery (Fig. 1A). After release, vesicles are taken back up through endocytosis, reloaded with neurotransmitters, docked to the membrane, and finally primed for repeated release. The dynamics of these recovery processes have been measured using a variety of experimental setups that elucidated their molecular pathways, their speed, and the numbers of vesicles involved[1,2]. Yet, it remains poorly understood how these individual stages of recovery interact, and ultimately affect synaptic signal transmission and synaptic computation.

An important feature of these recovery processes is that they operate on vastly different timescales, and thus might affect synaptic recovery in different ways. These timescales range from the orders of 100 ms (vesicle priming), over 1 s (docking and loading), to 10 s (endocytosis). While synapses typically recover within seconds[3], previous research has shown that synaptic recovery can slow down under strong stimulation[4,5]. For example, in hippocampal neurons, this slow form of recovery can take tens of seconds, up to minutes[5]. One explanation of this slowing down of recovery is the depletion of vesicles that are available for release and a fallback on the slowest recovery process (endocytosis)[4]. Thus, there is evidence that both fast and slow recovery timescales play a role under different conditions.

In contrast to these findings, models that investigated what specific functions could be performed by synaptic adaptation typically employed a simplified synaptic model, with one recovery timescale (and, depending on the model, one timescale of facilitation)[6–9]. This simplification of the recovery dynamics allowed for important insights into how synaptic depression might shape neural computations, for example, through temporal filtering of spike trains, adaptation, and gain control[10]. Yet, the presence of processes with timescales across several orders of magnitude in the recovery pathway could lead to different implications for synaptic function, which are not captured by these simplified concepts.

One challenge in achieving an understanding of the functional aspects of multiple timescales in vesicle recovery is that the dynamics of their interaction are not well explored. Many previous studies were set up such that one or two timescales of synaptic recovery could be measured[3,4,11,12], and, consequently, vesicle pool dynamics were typically modeled with only (up to) two recovery timescales[4]. However, some evidence for more than two relevant timescales of synaptic recovery exists[13], which could point to the existence of more relevant timescales of vesicle pool recovery. Furthermore, it is not clear whether the parameters of the vesicle dynamics, which were measured in isolation[14–16], are consistent with the overall recovery dynamics that can be measured experimentally.

To address the question of whether the diverse timescales of different vesicle recovery steps play a substantial role in synaptic transmission, we

[1]Max-Planck-Institute for Dynamics and Self-Organization, Göttingen, Germany. [2]University Medical Center Göttingen, Institute for Neuro- and Sensory Physiology, Göttingen, Germany. [3]International Max Planck Research School for Neuroscience, Göttingen, Germany. [4]Institute for the Dynamics of Complex Systems, University of Göttingen, Göttingen, Germany. [5]Biostructural Imaging of Neurodegeneration (BIN) Center, Göttingen, Germany. [6]Excellence Cluster Multiscale Bioimaging, Göttingen, Germany. ✉e-mail: viola.priesemann@ds.mpg.de

**Fig. 1 | Model of the presynaptic vesicle cycle.**
**A** Model of the presynaptic vesicle cycle, as proposed in[14]. Vesicles transition stochastically from pool to pool with a given timescale. Timescales of transitions are taken from experimental measurements[14] (see SI) and become orders of magnitudes larger the further they are away from the release stage (Primed pool). Similarly, the maximum number of vesicles in pools (numbers in circles) is larger for pools farther away from release. Upon an input spike, vesicles in the primed pool are emitted with probability $p_{\text{fuse}}$. Subsequently, vesicles can age with probability $p_{\text{age}}$ and are dynamically resupplied if the synapse lacks vesicles. Since the ageing and supply processes are very slow (order of hours) they have limited significance for the analysis in this work. **B**, **C** (top) Example of how spikes (red) result in the release of vesicles (blue) and (bottom) how vesicle pool levels change over time. Pools close to release operate on a fast timescale (**B**), whereas pools far away operate on slow timescales (**C**).

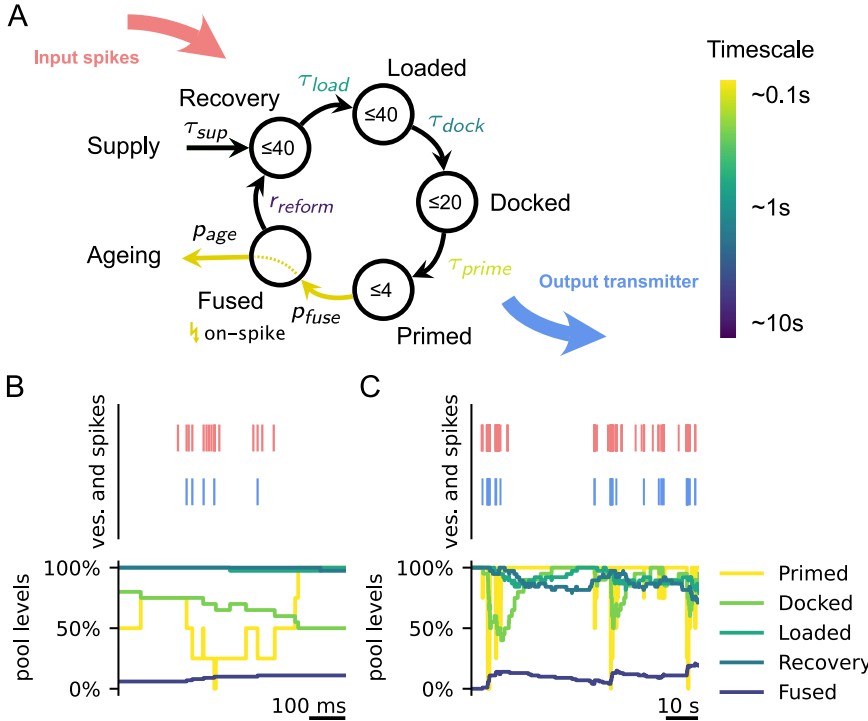

employ here both theoretical modeling and an analysis of cultured neurons. First, we analyze a multistage model of synaptic recovery in hippocampal synapses, which is based on timescales of recovery processes that were measured previously[14]. We show that, in this model, each of the timescales plays a role in shaping the overall recovery from release, leading to multi-timescale adaptation dynamics. We furthermore show that this form of depression can be described by a simplified synaptic model with effective 'power-law' adaptation. We then employ an experimental setup that allows us to measure vesicle release in hundreds of synapses in parallel. With this setup, we verify that the measured parameters in our model are close to the parameters that fit the experimental data.

Finally, to elucidate how power-law adaptation could contribute to neural information processing, we show that it specifically tunes down synaptic responses to strong, slow frequency fluctuations of presynaptic spiking. Our results suggest that hippocampal synapses are designed to efficiently transmit hippocampal activity, where such long timescale dependencies can be found[17,18].

## Results

### Multi-stage model of synaptic recovery
To simulate the presynaptic vesicle cycle, we employed the model proposed in[14]. This model incorporates the five most prominent stages of vesicle recycling in the presynapse as discrete vesicle pools: Previously released vesicles are recovered by endocytosis, loaded with neurotransmitters, docked to the membrane, and ultimately primed for release (Fig. 1A). Upon a spike, primed vesicles are released with a certain probability $p_{\text{fuse}}$ (in the main results $p_{\text{fuse}} = 0.2$). In between spikes, vesicles transition stochastically from pool $i$ to pool $j$ (e.g., docked to primed) with a filling level-dependent transition rate

$$r_{ij} = \frac{c}{\tau_{ij}}\frac{P_i}{P_i^{\max}}\left(1 - \frac{P_j}{P_j^{\max}}\right), \qquad (1)$$

where $\tau_{ij}$ is the timescale of vesicle transitions (e.g., $\tau_{prime}$), $c = 4$ ensures that $r_{ij} = 1/\tau_{ij}$ if pools are half full, and $P_i$ is the filling level and $P_i^{\max}$ the maximum capacity of pool $i$. Exceptions are vesicle priming, where $c = 2$, and the transition from the fused to the recovery pool via endocytosis, which happens with a fixed rate $r_{\text{reform}}$ as long as the fused pool is not empty and the

recovery pool is not full. Note, that there might be variation in the rate of endocytosis depending on the mechanism involved or the precise stimulation, for example the presence of bulk endocytosis under strong stimulation[2]. However, the extent of this variation under physiological conditions is not well known and we therefore model the average endocytosis rate as measured before[14]. The validity of this approach relies on the assumption that under the moderate stimulation we will use in our simulations, and due to the generally relatively slow rate of endocytosis, these effects play a limited role in determining synaptic coding function.

Intriguingly, the parameters of this model, which have been determined experimentally from cultured hippocampal neurons[14], have a very particular ordering on the cycle (Fig. 1A). Transition timescales between pools tend to increase with distance (i.e., number of steps on the vesicle cycle) from release. Vesicle priming can be as fast as 50 ms, while vesicle endocytosis and reformation takes several seconds for a single vesicle. Similarly, the upper limit for the sizes of pools increases with the distance to release. Since the effective recovery timescale of a pool depends on the combination of single vesicle recovery timescales and the pool size limit, these parameters result in very different effective recovery speeds of vesicle pools. Pools close to release recover on fast timescales in the order of 100 ms, whereas pools further away operate on timescales as slow as 10 s up to minutes (Fig. 1B, C).

### Power-law adaptation in the vesicle cycle
To investigate whether adaptation is dominated by one of these recovery timescales, or if all timescales play a role in adaptation, we set up several simulation experiments. First, we tested how the model synapse depressed under constant strong stimulation. We stimulated the model with a regular 50 Hz spike train (Fig. 2A) and measured how the average number of released vesicles per spike changes over time. To observe these changes on both short and long timescales, averages were taken over spikes grouped in log-spaced intervals. We found that the average of released vesicles decreased rapidly at the beginning of stimulation, but later this decrease slowed down and lasted up until about 100 s (Figs. 2B, S2, S3). In comparison, models with a single timescale of vesicle recovery showed only a very short phase of adaptation, after which vesicle release stayed constant. This indicates that, in our model of vesicle recovery, processes operating on long, medium, and short timescales interact to shape synaptic depression.

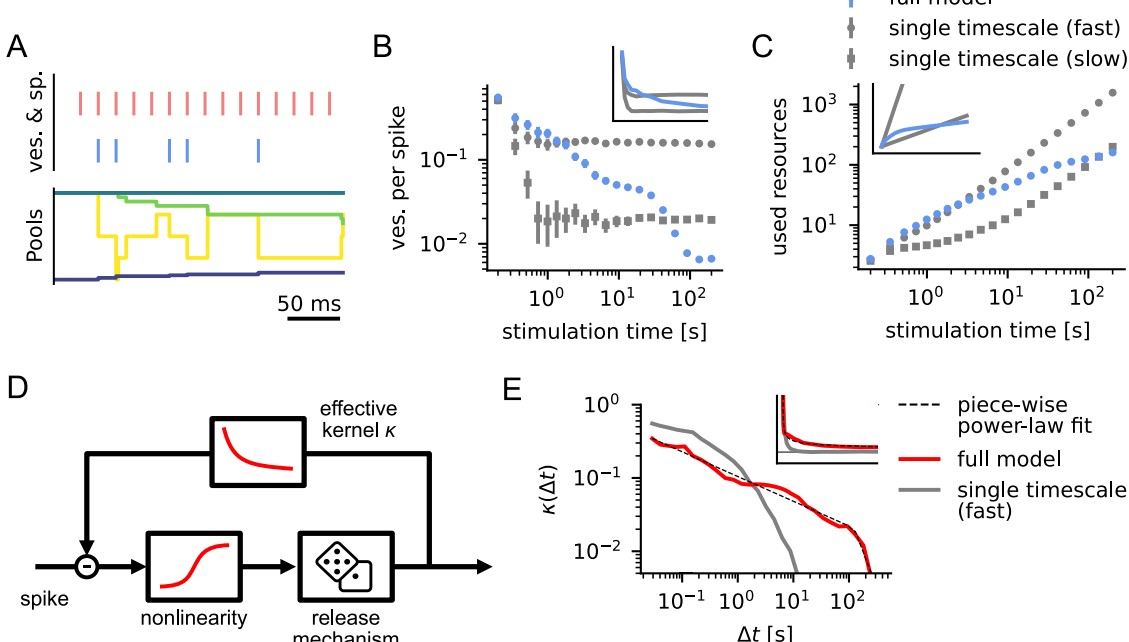

**Fig. 2 | Power-law adaptation resulting from multi timescale dynamics of the presynaptic vesicle cycle. A** Reaction of the vesicle cycle when stimulated with a 50 Hz signal (Color code as in Fig. 1B, C). **B** The average number of vesicles emitted per spike decreases approximately linearly in log-log scale up until ~10 – 100 s, which corresponds to a power-law decay. In comparison, models with a single timescale of vesicle recovery show a rapid decay that levels out after 1 s. Inset shows the data in linear scale from 0 to 5 s. Error bars denote 95% bootstrapping confidence intervals estimated from 100 experiments; inset shows the same data but linear axes (also in **C**, **E**). **C** The resources used (average number of vesicles emitted up to that time) increase rapidly for the single timescale model with fast recovery and slowly for the single timescale model with slow recovery. The full model strikes a balance between coding fidelity on short timescales and resource efficiency for strong stimulation on long timescales. **D** We used a simple response model to fit the full vesicle cycle model. An effective kernel $\kappa$ mediates negative feedback on the release probability after a vesicle release. **E** When fitted to responses to simulated spike trains (see Methods) the kernel can be approximated well by a power law $\kappa(\Delta t) \propto \Delta t^\alpha$ with exponent $\alpha \approx -0.3$ up to ~120 s (demonstrated here by fitting a piecewise-linear function in the log-log plot). This means that a single vesicle release has a measurable effect on future releases more than 2 minutes into the future. For the single timescale model with fast recovery (gray), this decay is much more rapid. The inset shows the kernels in linear scale up to $\Delta t = 80$ s. Kernel bins for very short and long $\Delta t$ that are difficult to estimate from the data have been excluded in this plot.

To reach a better understanding of how the release of a vesicle impacts the probability of release in the future, we fitted a generalized linear model (**GLM**) to our multi-stage model of vesicle recovery (Fig. 2D, see Methods). The GLM is an effective model of vesicle release, in which every spike triggers a vesicle release with a certain probability, but previous releases reduce this probability depending on an effective kernel $\kappa(\Delta t)$ (given the release was $\Delta t$ in the past). This adaptation kernel thus captures how releases reduce the future release probability on average. For example, a model similar to the classical Tsodyks-Markram model[19,20] without facilitation would employ an exponential kernel $\kappa(\Delta t) \propto \exp(-\Delta t/\tau)$. To estimate $\kappa(\Delta t)$, we stimulated the vesicle cycle model with highly variable spike trains ($\approx 0.5$ Hz with 1/f spectral power, see Methods) and measured the resulting releases. We then fitted this input-output behavior with the GLM using maximum likelihood estimation (SI Fig. S16), which we validated by comparing the fit to data on a test set (Fig. S5G, S16). The resulting effective kernel $\kappa(\Delta t)$ showed that previous vesicle releases have a measurable effect on release probability on a wide range of timescales, and could be well approximated by a power law $\kappa(\Delta t) \propto \Delta t^{-0.3}$, up to about 120 s (Fig. 2E). This power-law description of the kernel fits better than several other long-tail (or double exponential) candidate functions (SI, Fig. S15).

**Experimental validation of model dynamics**

Our simulation experiments indicated that, under strong stimulation, vesicle releases can significantly impact the efficacy of the synapse, even minutes into the future. To test this prediction, we set up an in-vitro experiment with rat hippocampal cultures to measure the long timescale dependence of synaptic exhaustion on vesicle releases. In our experimental protocol we first aimed to partially exhaust synaptic vesicle pools by strong stimulation, let the synapse recover while pausing stimulation, and then tested the final efficacy of the synapse (Fig. 3A). By varying the time of exhaustion and pausing over several orders of magnitude, we could then get an understanding of the dependence of synaptic depression over a range of timescales (Figs. 3B, S6). To further constrain our model, in a second experiment we also measured the cumulative number of vesicle releases in response to a constant stimulation of 5 Hz (Fig. 3G).

As a measure of synaptic release, we turned to super-ecliptic pHluorin, a pH-sensitive fluorophore that is introduced within the lumen of synaptic vesicles, where it is quenched by the acidic milieu, and reports their exocytosis by un-quenching upon reaching the neutral extracellular buffer[21]. Typical pHluorin experiments involve the expression of synaptic vesicle proteins that carry this molecule on their intravesicular chains. To avoid any effects of genetic manipulation and protein expression, we turned to antibodies against the intravesicular domain of synaptotagmin 1, which carried secondary nanobodies[22] conjugated to pHluorin molecules[23]. The synaptotagmin antibodies were loaded within all of the active vesicles, by prolonged incubations, before the analysis[24]. This procedure enables the analysis of synaptic release dynamics and should provide virtually identical results to conventional pHluorin measurements[25]. To measure the cumulative number of released vesicles in the second experiment, we prevented the reacidification of vesicles after reuptake, by incubating the neurons with bafilomycin[26].

Our experiments indeed showed that synaptic depression is captured well by the model (Fig. 3E). As predicted from our model, the speed of recovery changed based on the duration of stimulation, with long stimulation drastically slowing down the recovery. This behavior could not be reproduced by a model with single timescale recovery (Fig. 3F). Furthermore, the cumulative response of the synapse to constant stimulation starts to plateau after 1 min of stimulation, after most vesicles have been released

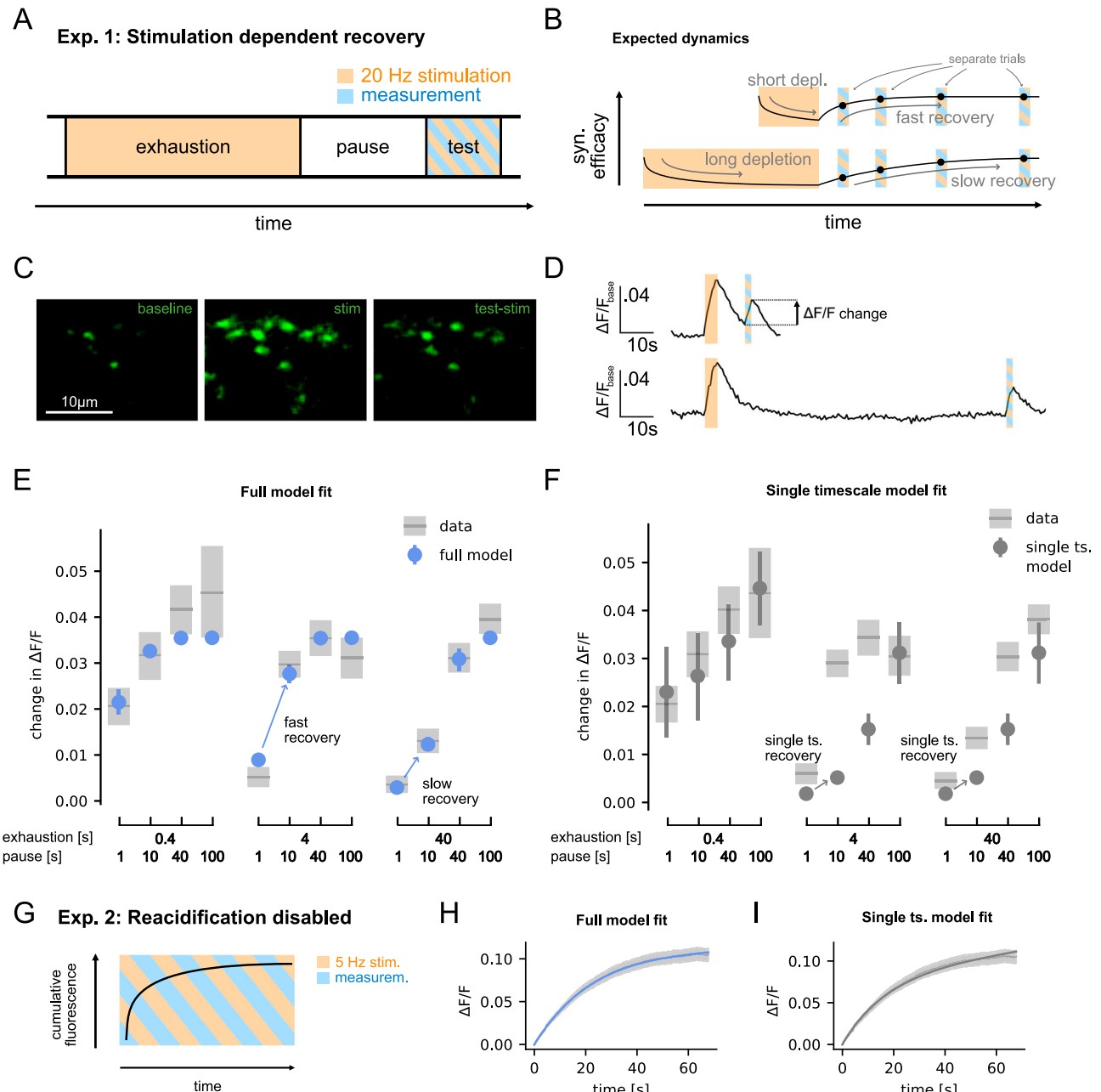

**Fig. 3 | Hippocampal synapses in vitro show stimulus-dependent recovery timescales. A** Cultured synapses were first exhausted using 20 Hz stimulation, recovered for a variable time, after which the synaptic efficacy was tested. **B** From our model, we expected that longer exhaustion times result in longer recovery timescales. The conditions with different exhaustion and pause time were conducted with separate coverslips without repetitions. **C** Fluorescence images of synaptic activity recorded with super-ecliptic pHluorin during baseline activity (left), strong stimulation during the exhaustion period (middle) and test stimulation in the test period (right). For display purposes, the images were deconvolved using the Richardson-Lucy algorithm from the DeconvolutionLab2[67]. **D** Example fluorescence traces recorded from a single region of interest as an average across multiple synapses for an exhaustion time of 4 s, and a pause time of 10 s (top) or 100 s (bottom). **E** The

recordings indeed show that longer exhaustion time leads to slower recovery. This can be fitted well by the full vesicle cycle model. **F** A single timescale recovery model cannot fit this difference. The effective recovery timescale of the fitted model is ≈120 s, which does not change depending on the exhaustion time. **G** With reacidification disabled, fluorescence of released vesicles accumulates under constant stimulation. **H** The full model is able to capture how the fluorescence plateaus. **I** Also the single timescale model can fit this data reasonably well, but it requires extremely low release probabilities ($p_{fuse}$ ≈0.01) and slow recovery rates to achieve this (SI Fig. S8). **E, F, H, I** For experiments, shaded areas denote 95% bootstrapping confidence intervals; for simulations, bars/shaded areas denote 95% confidence intervals based on the posterior distribution of the Bayesian models.

once, which is well captured by the full model (Fig. 3H). Also the single timescale model can fit this data reasonably well (Fig. 3I), but it requires extremely low release probabilities ($p_{fuse}$ ≈ 0.01) and slow recovery rates to achieve this (SI Fig. S8). We showed this by fitting the model parameters to the data of both experiments simultaneously using Bayesian inference with MCMC (see Methods). Parameters of the full model were well constrained

by priors that we formulated based on previous experimental measurements (see SI Fig. S7). All fitted parameters were shared for both datasets, except for the release probability $p_{fuse}$, which had to be fitted independently to properly account for the data. We assume that this difference accounts for the difference in presynaptic Calcium levels under different stimulation rates, which change baseline vesicle release probability[27]. Indeed, as this would

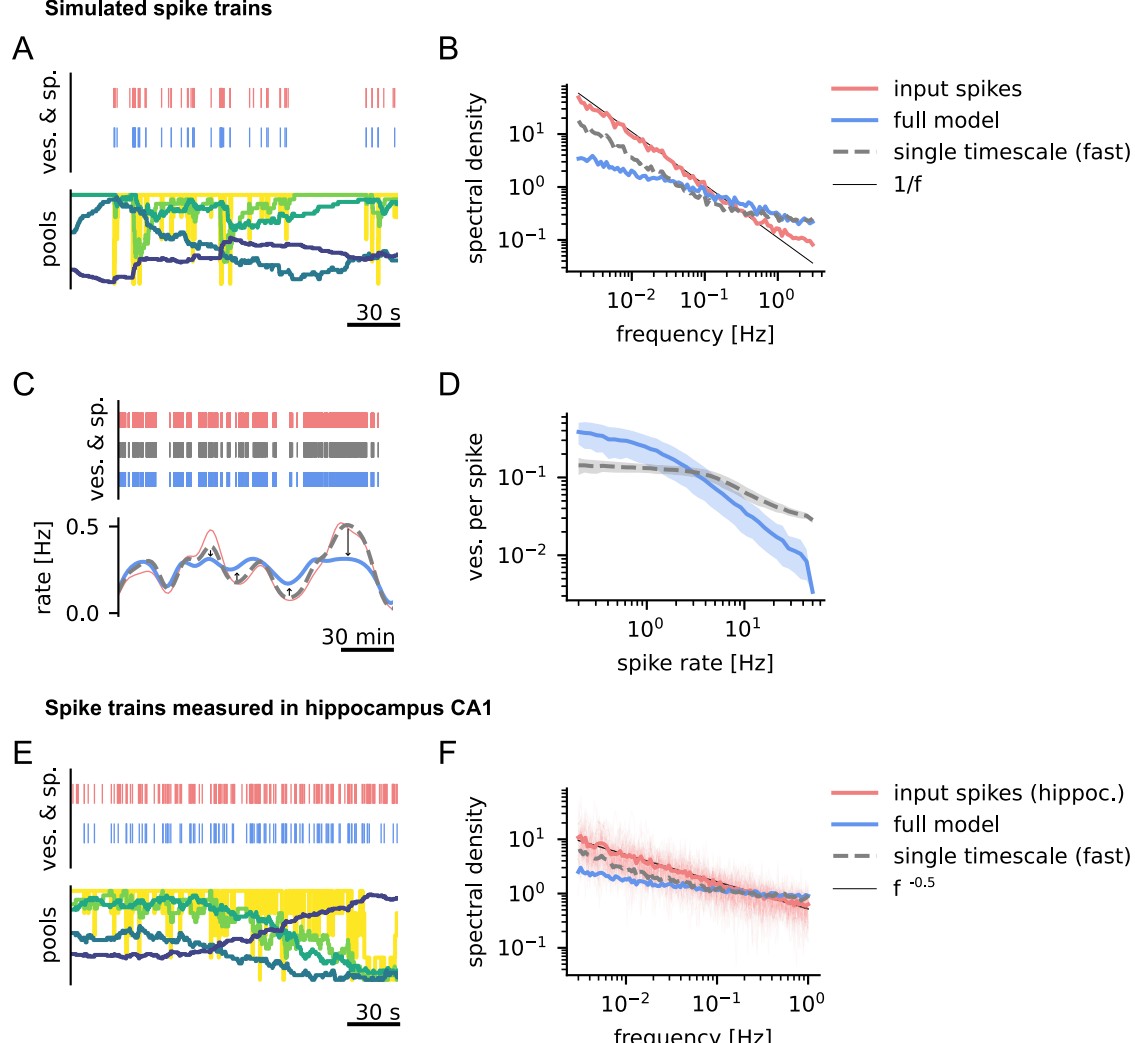

**Fig. 4 | Power-law adaptation enables temporal whitening of hippocampal spike trains. A** Reaction of the vesicle cycle when stimulated with a spike train with 1/f power spectrum and an average rate of 1 Hz (Color code as in Fig. 1B, C). **B** The response of the vesicle cycle model shows a flatter spectrum when compared to the original 1/f spectrum of the input. In comparison, the model with a single vesicle recovery timescale only attenuates high-frequency fluctuations. Power spectra are plotted as normalized densities. The full and single timescale models have comparable average power, with average release rates of 0.26 Hz and 0.29 Hz, respectively. **C** Power-law adaptation dynamically adjusts the releases per spike to the input rate. Here, we demonstrate this behavior in the strong input regime (~4 Hz), which allows for clear visualization. During high activity, releases per spike are selectively reduced, mainly mediated by slow endocytosis. During low activity, they are enhanced when compared to a single timescale model with similar average release rate. The single timescale model closely follows the linear prediction from input spikes (thin red line). **D** This can be seen more clearly when computing the average number of vesicles released per spike depending on the input spike rate. **C**, **D** Rates were estimated by convolving the spike and release trains with kernels (see Methods). **E** Reaction of the vesicle cycle when stimulated with a spike train recorded from hippocampus CA1[62]. **F** Hippocampal spike trains show an approximate $f^{-0.5}$ power spectrum over the relevant range. Here the full model nearly flattens the spectrum. Spectra for individual spike trains (transparent lines) behave similarly to their mean, showing that the power law is not a result of averaging spectra of spike trains with different single timescales[68].

suggest, the estimated $p_{fuse}$ is small for slow stimulation ($p_{fuse} \approx 0.08$ for 5 Hz) and large for fast stimulation ($p_{fuse} \approx 0.7$ for 20 Hz).

### Temporal whitening of hippocampal spike trains through power-law adaptation

Finally, we considered what functional advantage power-law adaptation in synapses could bring for synaptic information transmission. To increase the efficiency of neural communication, it has long been proposed that redundant information should be discarded, which could be implemented by synaptic adaptation[7]. Natural signals often contain correlations on many timescales, in which case redundancy reduction, i.e., neural adaptation, should also operate on many timescales[28]. Based on our model, we hypothesized that hippocampal synapses could perform such a redundancy

reduction in the longer timescale regime, i.e., from several hundreds of milliseconds to several minutes.

Formally, for temporal signals, the goal of efficient coding is to reduce the auto-correlation in the encoded signal. This is commonly illustrated by showing that the encoding flattens the power spectral density (**PSD**) of the signal (i.e., achieving the PSD of a temporally uncorrelated white noise signal, which is also called temporal whitening)[28]. To test whether the vesicle cycle model could implement this coding strategy, we stimulated it with simulated spike trains exhibiting correlations on long and short timescales, that is, spike trains with 1/f PSD (Fig. 4A, see Methods). Note, that perfect 1/f scaling over all frequencies is not achievable with a realistic spike train, since it has a finite number of spikes. We therefore generated a signal with 1/f scaling in the longer timescale regime where we considered synaptic

whitening to be most relevant. Our simulations showed that the depression dynamics of the full vesicle cycle model partially flattened the PSD of the signal, meaning it partially removed correlations over several orders of magnitude of timescales (Figs. 4B, S1). In comparison, the single timescale vesicle cycle model only flattened the PSD for high frequencies. One intuitive way to understand whitening is that it dampens fluctuations in the input rate on slow timescales, which can be extremely large (Fig. 4C). During periods of high activity, where spikes are very predictable, average releases per spike are dynamically tuned down, which leaves more resources for periods of low activity, with less predictable spikes (Fig. 4C). Thus, power-law adaptation in the presynaptic vesicle cycle could enable efficient synaptic information transmission for spike trains with long-tail auto-correlation functions.

Based on these results we wondered whether this form of efficient coding could be relevant in the rat hippocampus. To test this, we analyzed long recordings of CA1 activity of freely behaving rats (see Methods), and simulated the responses of the vesicle cycle under this stimulation (Fig. 4E, F). We observed that hippocampal spike trains have approximately $f^{-0.5}$ PSD in the longer timescale regime, which, similarly to $1/f$ PSD, is associated with long tail autocorrelations[29]. In comparison, the vesicle release train generated by the response of the vesicle cycle model is nearly flat (Fig. 4F). This effect can also be seen in the measured coefficient of variation (**CV**) of the activities (where CV=1 is associated with Poisson activity and maximum information rate). Measured spike trains had a mean CV of 3.4 (1.4 to 15), while simulated releases had mean CVs of 1.8 (0.9 to 3.3) and 2.7 (1.6 to 7.7) for the full and the single timescale model, respectively. Overall, this shows that, not only is the vesicle cycle able to remove long tail correlations from the input spike trains, but it is also approximately tuned to the type of correlations found in hippocampal activity.

## Discussion

Here we analyzed a model of synaptic vesicle release based on experimentally determined parameters of the most prominent steps of the vesicle recycling machinery, which allowed us to generate large amounts of data for the input-output function of the presynapse. Our analysis showed that the interplay of the different recycling steps can lead to multi-timescale adaptation, which implies a slowdown of recovery under continued stimulation. These results motivated us to specifically test for the presence of both, short and long timescales of synaptic recovery in cultured hippocampal neurons. Our experiments demonstrated that synaptic recovery can indeed operate on a range of timescales, even up to minutes, depending on the stimulation protocol. Finally, we showed that synaptic multi-timescale adaptation is effective in attenuating high-amplitude, low-frequency fluctuations in hippocampal spike trains, which enables an efficient allocation of synaptic resources for synaptic transmission.

The multi-timescale depression we described is a result of the particular ordering of vesicle pool sizes and timescales in the vesicle cycle. Under continued stimulation, smaller vesicle pools close to the release site are depleted first, whereas larger pools further away can continue to supply vesicles for a longer time. This leads to a gradual shift of the recovery bottleneck from later (faster) to earlier (slower) stages of recovery (Figs. 2, SI Fig. S5). The interaction of these processes in our model leads to effective depression and recovery timescales up to the order of minutes (Fig. 2B, E)—nearly an order of magnitude slower than the slowest recovery rate in the model. Under moderate synaptic stimulation this recovery can be described by a generalized linear model with an effective power-law depression kernel up to about 120 s (Fig. 2E). A shortcoming of this linear description is that it does not capture the deep depression of the synapse under very strong stimulation, which can occur in the vesicle pool model (SI Fig. S5). Nevertheless, for the functionally more relevant moderate stimulation regime, the power-law adaptation model yields a simplified description of the cascading dynamics of vesicle pool recovery.

Previously, several proposals have been made to explain why recovery slows down under continued stimulation. Candidate mechanisms include the long-lasting depletion of vesicle pools[3,4,30], the blocking of release sites[5],

or the replacement of tethered vesicles on the presynaptic membrane[31]. Our results corroborate the idea that the depletion of vesicle pools is a central limiting factor of synapses under continued stimulation. Vesicle cycle parameters only had to be slightly tuned to fit the experimental data (SI Fig. S7), meaning that the timescales and sizes of vesicle pools that were measured previously yield a plausible explanation of the overall timescales of synaptic recovery.

In comparison to the full vesicle cycle model, a single timescale recovery model was not able to fit the experimental data as well (Fig. 3). This is also the case when comparing the models using Bayesian model comparison, which takes the number of fitted parameters of the models into account (specifically, leave-one-out cross validation and the widely applicable information criterion both clearly favor the full model, see SI Fig. S11). Bayesian model comparison also favors the full model (as well as a power-law GLM, Fig. S10) over a model with two timescales of vesicle recovery (SI Fig. S11). While the two timescales model can succeed in modeling a change of recovery timescale depending on the stimulation, it failed to describe the data from both experiments simultaneously as well as the full model (SI Fig. S9, S11). It seems that there are different requirements between the multi timescale depression (Fig. 3E) and the need to release enough vesicles to saturate the fluorescence signal (Fig. 3H), that lead to contradictions within the single and two-timescale models. Still, while our experiments test recovery on medium and longer timescales, they do provide little data on synaptic recovery on faster timescales, which are predicted by our model. Although these shorter timescales have been found in other experiments[11,13,32], a full experimental validation of the model would require a different experimental setup that can measure very short, medium, and long timescales of synaptic recovery simultaneously. Surprisingly, however, when fitting the full model without precise priors on the recovery timescales, the model recovers all timescales of recovery from the data alone (SI Fig. S14), which constitutes an indirect validation of also the faster recovery dynamics.

Synaptic transmission is the most energy-intensive process in neural computation and is estimated to consume about 40% of the total cerebral energy budget[33,34]. Consequently, synapses face a strong energy/information trade-off, which often means that synapses should actively reduce the number of uninformative releases. Previous studies have shown that the depression of synaptic release can improve the amount of information transmitted per release[6–8,35,36], by dynamically lowering synaptic responsiveness when current spikes can be predicted from past spikes. This is a general mechanism that enables the efficient encoding of events, which, for example, also applies to neural spike-frequency adaptation[28,37]. Our results extend these previous ideas and suggest that synaptic depression improves information transfer not only on fast timescales, as suggested before[6–8,35,36], but rather on a broad range of fast to slow timescales.

More specifically, we showed that power-law adaptation is effective in decorrelating synaptic transmission events, even when driven by input spike trains with correlations on several orders of magnitude (Fig. 4). During extended periods of high activity, where spikes become highly predictable, power-law adaptation dynamically tunes down synaptic responsiveness in order to conserve energy (Figs. 2C, D, 4C). A similar attenuation of slow rate fluctuations is observed in a wide range of systems in the form of firing-rate adaptation[38–40]. An interesting feature of this adaptation is that it changes dynamically depending on the timescale of fluctuations in the stimulus history. Here, we found a surprisingly simple mechanism for history-dependent adaptation in the synaptic vesicle cycle: The existence of multiple stages of recovery leads to a history-dependent depletion of vesicle pools, which tunes the timescale of adaptation to the timescale of rate fluctuations in the input spike train (SI Fig. S17).

Not all spike trains, however, are encoded efficiently by power-law adaptation, and we therefore wondered whether this adaptation is matched to the correlations found in hippocampus. Consider, for example, a (hypothetical) neural code that consists of clusters of spikes separated by intervals of silence with unpredictable length, which only has correlations on the length of the clusters. In this case, synaptic adaptation should only operate on the timescale of clusters, but not longer timescales, as this would

reduce responsiveness to the unpredicted (not transmitted) events[7]. In contrast, we found that spike trains measured in the rat hippocampus (CA1) show slow rate fluctuations that follow a power-law over almost three orders of magnitude (similar to findings of previous research[17,18], SI Fig. S15). These fluctuations are whitened by encoding them via multi-stage recovery, which means that the strong correlations on long timescales are effectively removed in the encoded signal (Fig. 4F). This indicates that synaptic power-law adaptation is tuned to the statistics of spike trains in hippocampus, and thus could be an important contribution to its efficient functioning.

Our model focused on the dynamics of vesicle recovery, although other processes are also affecting presynaptic adaptation. Most importantly, this includes the presynaptic calcium dynamics, which effect synaptic facilitation and the speeding-up of vesicle recovery, but possibly also other factors[3,27]. These processes critically determine how synapses transmit signals on shorter timescales, and thus should become important when considering information transmission on a finer temporal scale than we did here (although there are some exceptions[41]). For example, naive temporal whitening is only useful for high signal-to-noise ratios, where redundancy reduction directly leads to an efficient code. However, in the face of a more noisy signal, an optimal signal encoding would require a more robust code that exploits redundancies[37,42,43]. At the synapse, such a robust code (or other types of codes) could for example be achieved through short-term facilitation[44], or different, activity dependent endocytosis mechanisms[2], which we did not model here. With a better understanding of these processes, it will be important to consider how they can interact with multi-stage vesicle dynamics to shape synaptic function.

Beyond the efficient transmission of information, synaptic adaptation has also been suggested to have other important purposes for neural computations. For example, neural adaptation could allow to transmit only certain aspects of multiplexed neural codes[45]. On the network level, synaptic adaptation can take a central role in shaping dynamics, for example through the induction of synchronization[46], the alteration of attractor dynamics[47–49], or memory effects[50,51]. Especially on the network level, power-law adaptation could have a wide range of important implications that extend these previous concepts, for example by contributing to homeostatic regulation across a wide range of activity levels[52].

Multi-stage vesicle recovery is a plausible mechanism for long timescales of synaptic depression. Aside from efficiently transmitting signals, long timescales of synaptic adaptation could have a wide range of implications for neural function, especially on the network level. The effective description of multi-stage vesicle recovery through a power-law adaptation kernel will allow to test this in large-scale simulations.

Our results also raise the question of whether synapses could be tuned to their presynaptic firing statistics more generally. For example, hair cell synapses, which code for auditory stimuli that commonly have 1/f spectrum[53], have been implicated to employ power-law adaptation[54], and some cortical synapses are also known to possess at least two timescales of adaptation[11]. Given the variety of functional subtypes of synaptic dynamics[55], which can depend systematically on the spikes they encode[56], it would be interesting to investigate the relationship between spiking statistics and the temporal dynamics of adaptation in the corresponding synapses—not only in hippocampus but also in other neural subsystems.

## Methods
### Simplified response model of vesicle cycle
To obtain a simplified description of the vesicle cycle, we modeled the input-output response using a generalized linear model (GLM), similar to what has been used to model neural responses[57]. In this model, every incoming spike has the chance to trigger a release event of a vesicle ($r(t) = 1$), or not ($r(t) = 0$), with a probability that depends on a base release probability, and a reduction based on past releases

$$p\left(r(t) | \overrightarrow{t}_{\text{past}}, \theta\right) = \frac{e^{r(t)a(t)}}{1 + e^{a(t)}}, \quad (2)$$

$$a(t) = b - \sum_{t' \in \overrightarrow{t}_{\text{past}}} r(t')\kappa(t - t'). \quad (3)$$

Since the synapse can release multiple vesicles, we assume that the total number of releases $k(t)$ can be computed via the binomial distribution

$$p(k(t)|q(t)) = \binom{N}{k} q^k (1 - q)^{N-k}, \quad (4)$$

where $N = P_{\text{primed}}^{\max} = 4$, and for readability we define $q(t) = p(r(t)|\overrightarrow{t}_{\text{past}}, \theta)$ and dropped the time dependency.

We parameterized the kernel $\kappa$ through 40 values in log-spaced bins (with boundaries $\{e^q : q \in \{\log(a_{\min}) + j/39(\log(a_{\max}) - \log(a_{\min})) : j \in [0..39]\}\}$, where $a_{\min} = 0.001$ s, $a_{\max} = 500$ s). Maximum-likelihood estimates for parameters $\theta = \{\hat{b}, \hat{\kappa}\}$ that describe the simulated data (vesicle releases in response to 1/f spike-trains) were then obtained via gradient ascent on the log-likelihood (SI Fig. S16). Similar models of synaptic recovery have been employed before[11,13], which however used a set of exponential functions instead of bins to parameterize the kernel.

### Bayesian modeling of experimental data
Parameters of the models were fitted to the experimental data using Bayesian inference with MCMC sampling. To make sampling tractable, we first switched the stochastic description of vesicle dynamics to a deterministic description. This amounts to simply replacing stochastic vesicle transitions between pools through deterministic differential equations, such that for pool $P_i$

$$\frac{\partial P_i}{\partial t} = r_{i-1,i} - r_{i,i+1}, \quad (5)$$

where $r_{ij}$ are transition rates from pools $i$ to $j$ as before. Similarly, vesicle release was replaced by a continuous release rate $r_{\text{primed,fused}} = p_{\text{fuse}} r_{\text{stim}} P_{\text{primed}}$, where $r_{\text{stim}}$ is the stimulation rate given by presynaptic spiking. This model shows comparable dynamics to the stochastic model (Fig. S5).

Using MCMC, we then estimated the model parameters $\theta = \{\alpha_{\text{obs}}^{(1)}, \alpha_{\text{obs}}^{(2)}, \sigma_{\text{obs}}^{(1)}, \sigma_{\text{obs}}^{(2)}, \sigma_{\text{obs}}\} \cup \theta_{\text{obs}}$, where the $\alpha_{\text{obs}}$ are factors that scale the number of released vesicles (model) to the increase in dF/F (experiment), $\sigma_{\text{obs}}$ are the model errors (see below), and $\theta_{\text{obs}} = \{p_{\text{fuse}}^{(1)}, p_{\text{fuse}}^{(2)}, r_{ij}, P_i^{\max} : i, j \in \text{Pools}\}$ are the vesicle cycle parameters. Here, the $\alpha_{\text{obs}}$ and $p_{\text{fuse}}$ were used to model experiment 1 or 2 (Fig. 3), respectively. Note, that we employ a hierarchical prior $\sigma_{\text{obs}}$ for the model errors $\sigma_{\text{obs}}^{(1)}$ and $\sigma_{\text{obs}}^{(2)}$ (for details see SI table 1), since we expect both experiments to be fitted similarly well by the models, and as a way to prevent any experiment from dominating the likelihood with a very small inferred error. Also note, that the width of the likelihood will also depend on the empirical measurement error, as explained below. For computation, we employed the Python package PyMC3[58] with NUTS (No-U-Turn Sampling)[59] using multiple, independent Markov chains.

Formally, we computed the log posterior probability of the parameters $\theta$ given the experimental data of both experiments $D^{(1)} = \{D_i : i \in \text{conditions}\}$, $D^{(2)} = \{D_t^{(2)} : t \in \{0..T\}\}$:

$$\log p(\theta | D^{(1)}, D^{(2)}) \propto \log p(D^{(1)}|\theta) + \log p(D^{(2)}|\theta) + \log p(\theta). \quad (6)$$

In experiment 1, the data $D_i^{(1)}$ are the recorded increases in fluorescence ($\Delta F/F$ change) for all synapses in one condition $i$, and the conditions are all tested combinations of exhaustion and pause time, i.e., conditions=$\{(T_{exh}, T_{pause}) : T_{exh} \in \{0.4\,\text{s}, 4\,\text{s}, 40\,\text{s}\}, T_{pause} \in \{1\,\text{s}, 10\,\text{s}, 40\,\text{s}, 100\,\text{s}\}\}$. Since each experimental condition was tested on independent synaptic samples (see Imaging), the likelihood was defined as a product of Gaussian

likelihoods for each condition

$$p(D^{(1)}|\theta) = \prod_{i\in\text{cond.}} \mathcal{N}(\bar{D}_i^{(1)}; \mu = \alpha_{\text{obs}}^{(1)} O_i(\theta_{\text{obs}}), \sigma = \sigma_{\text{obs}}^{(1)} + \text{SEM}(D_i^{(1)})),$$

(7)

where $\bar{D}_i^{(1)}$ is the empirical mean over recorded synapses, $\text{SEM}(D_i^{(1)})$ is the error of the mean, and $O_i(\theta_{\text{obs}})$ is the number of releases measured in the simulation under stimulation condition $i$ and parameters $\theta_{\text{obs}}$.

In experiment 2, the data $D_t^{(2)}$ are the fluorescence traces measured in 1231 synapses over time. Again, assuming independence of experimental noise, the likelihood was defined as a product of Gaussian likelihoods for each time point

$$p(D^{(2)}|\theta) = \prod_{t\in\{0..T\}} \mathcal{N}(\bar{D}_t^{(2)}; \mu = \alpha_{\text{obs}}^{(2)} O_t(\theta_{\text{obs}}), \sigma = \sigma_{\text{obs}}^{(2)} + \text{SEM}(D_t^{(2)})),$$

(8)

with most variables defined similar to experiment 1. $O_t(\theta_{\text{obs}}) := n_t$ is the estimated number of vesicles that have been released once (i.e., are fluorescent). Under the assumption that previously released and unreleased vesicles are well mixed, this number can be computed as $n_{t+1} = n_t + r_t f_t$, where $f_t = 1 - n_t/N$ is the fraction of unreleased vesicles, $N$ is the total number of vesicles, and $r_t$ is the number of vesicles emitted at $t$. This ensures that the model does not predict more fluorescent vesicles than there are total vesicles.

Details about the prior distributions $p(\theta)$ can be found in SI Table 1.

## Creation of 1/f spike trains

To simulate a point process exhibiting a 1/f power spectrum we employed the intermittent Poisson process proposed in[60], which allowed us to tune average firing rates and slopes of the power spectrum. The intermittent Poisson process models an emitter with 'on' and 'off' phases, where it generates bursts of events with a rate $1/\tau_0$ or is silent respectively. The exponent of the spectrum can be controlled by tuning the distribution of the length of 'on' and 'off' phases, which we chose such as to obtain a 1/f spectrum (for details see ref. 60). To tune the firing rate we scaled all emission times with a constant factor. 'On' phase burst rates of $1/\tau_0 = 25$ Hz, 37.5 Hz and 50 Hz resulted in total rates of the process of 0.5 Hz, 0.75 Hz and 1 Hz, respectively. See also the simulation code for details.

## Data of hippocampal neuron spiking

Estimating the power spectra of recorded spike trains requires long recordings under natural conditions. We here used the CRCNS hc-11 dataset[61,62], which is composed of eight bilateral silicon-probe multi-cellular electrophysiological recordings performed in CA1 of four freely behaving male Long-Evans rats. The recordings were performed during 6–8 h of rest/sleep time in cages, and 45 min of a maze running task, which we considered representative of typical rat behavior. We found that the power spectra exhibited a slight mean rate dependence, with low rate units having a flatter spectrum (SI Fig. S15). For the results in the main paper, we, therefore, excluded high-rate units (>5 Hz), which likely are interneurons or multi-units, and low-rate units (< 0.7 Hz), for which estimated power spectra become more difficult to estimate, and which we considered less relevant for efficient coding. Nevertheless, the results for low-rate units are qualitatively the same (SI Fig. S15).

## Estimation of power spectra

Power spectra of point processes can be estimated directly from the spike times, without binning the data. This brings the advantage that sample frequencies $\omega$ can in principle be chosen freely and are not constrained by the bin size or the length of the data, as they would be for the Fast Fourier Transform that is commonly used to compute the spectrum. Note that, for slower frequencies, bins should still be chosen as multiples of $1/T$, where $T$ is the data length, in order to avoid resonance effects in the estimation.

However, results in the main paper (Fig. 4) are in the frequency regime where these can be neglected (see SI Fig. S15).

An estimate for the power-spectrum $S[\omega]$ of the data (sampled from time $t = 0$ to $T$) can be obtained via[63]

$$S[\omega] = \left\langle \frac{1}{T} \left\| \sum_{t_{\text{event}}} \exp(-i2\pi\omega t_{\text{event}}) \right\|^2 \right\rangle,$$

(9)

where $\langle\cdot\rangle$ is an average over $n$ realizations. For simulation experiments (Fig. 4B) we chose $n = 100$ and $T \approx 1000$ h, computed $S[\omega]$ for $\omega$ in 100 log-spaced bins, and lastly normalized the integral of the obtained spectrum to 1. For experimental data (Fig. 4E) the same procedure was applied, but we split the data into parts of equal length before averaging, in order to reduce noise in the estimation. Thus, the number of realizations was $n = (\#\text{sorted units})(\#\text{splits})$, where $\#\text{sorted units} = 49$ and $\#\text{splits} = 5$, as well as $T \approx 6\text{h}/5 \sim 10\text{h}/5$, depending on the recording.

## Estimation of release rates

To illustrate the effect of power-law adaptation, we computed the dynamical release rate of the models (Fig. 4C, D). To this end we convolved the release trains $r(t) = \sum_{t_{\text{event}}} \delta(t - t_{\text{event}})$ with a kernel $k(t, t')$

$$\text{rate}(t) = \int_{t'=0}^{t_{\text{max}}} r(t')k(t, t')dt' = \sum_{t_{\text{event}}} k(t, t_{\text{event}}).$$

(10)

For illustration purposes, in Fig. 4C we chose a Gaussian kernel with width $\sigma = 300$ s. To compute the rate-rate plot in Fig. 4D, we used a window kernel $k(t, t') = 1/W$ if $|t - t'| < W/2$ else 0, with $W = 50$ s. Results do not critically depend on the widths of the kernels, although relatively wide kernels are required to show the effects of fluctuations on slow timescales.

## Animals

Animals were handled according to the regulations of the local authorities, the University of Göttingen, and the State of Lower Saxony (Landesamt für Verbraucherschutz, LAVES, Braunschweig, Germany). All animal experiments were approved by the local authority, the Lower Saxony State Office for Consumer Protection and Food Safety (Niedersächsisches Landesamt für Verbraucherschutz und Lebensmittelsicherheit), and performed in accordance with the European Communities Council Directive (2010/63/EU).

## Rat dissociated hippocampal cultures

Newborn rats (Rattus norvegicus) were used for the preparation of dissociated primary hippocampal cultures, following established procedures[64,65]. Shortly, hippocampi of newborn rat pups (wild-type, Wistar) were dissected in Hank's Buffered Salt Solution (HBSS, 140 mM NaCl, 5 mM KCl, 4 mM NaHCO₃, 6 mM glucose, 0.4 mM KH₂PO₄, and 0.3 mM Na₂HPO₄). Then the tissues were incubated for 60 min in enzyme solution (Dulbecco's Modified Eagle Medium (DMEM, #D5671, Sigma-Aldrich, Germany), containing 50 mM EDTA, 100 mM CaCl₂, 0.5 mg/mL cysteine, and 2.5 U/mL papain, saturated with carbogen for 10 min). Subsequently, the dissected hippocampi were incubated for 15 min in a deactivating solution (DMEM containing 0.2 mg/mL bovine serum albumin, 0.2 mg/mL trypsin inhibitor, and 5% fetal calf serum). The cells were then triturated and seeded on circular glass coverslips with a diameter of 18 mm at a density of about 80,000 cells per coverslip. Before seeding, all coverslips underwent treatment with nitric acid, sterilization, and coating overnight (ON) with 1 mg/mL poly-L-lysine. The neurons were allowed to adhere to the coverslips for 1 h to 4 h at 37 °C in plating medium (DMEM containing 10% horse serum, 2 mM glutamine, and 3.3 mM glucose). Subsequently, the medium was switched to Neurobasal-A medium (Life Technologies, Carlsbad, CA, USA) containing 2% B27 (Gibco, Thermo Fisher Scientific, USA) supplement, 1% GlutaMax (Gibco, Thermo Fisher Scientific, USA) and 0.2% penicillin/streptomycin mixture (Biozym Scientific, Germany).

The cultures were then incubated at 37 °C, and 5% $CO_2$ for 13–15 days before use. Percentages represent volume/volume.

### Labeling

Before labeling, the primary mouse anti-synaptotagmin1 antibody (Cat# 105 311, Synaptic Systems, Göttingen, Germany) at a dilution of 1:500 and the secondary anti-mouse nanobody, conjugated to superecliptic pHluorin (custom made, NanoTag, Göttingen, Germany) at a dilution of 1:250, were preincubated in neuronal culture medium (constituting 10% of the final volume for labeling) for 40 min at room temperature (RT). The preincubation was performed to ensure the formation of a stable complex between the primary antibody and the secondary nanobody. The labeling solution's volume was then increased to the final volume needed for the labeling procedure (300 $\mu$L per coverslip). Following a brief vortexing, 300 $\mu$L of labeling solution was pipetted to the wells of a new 12-well plate (Cat# 7696791, TheGeyer, Renningen, Germany). Subsequently, the coverslips were transferred to the well plate. The neuronal cultures were then incubated for 90 min at 37 °C. After incubation, the cell cultures were washed 3 times in pre-heated Tyrode's solution (containing 30 mM glucose, 5 mM KCl, 2 mM $CaCl_2$, 124 mM NaCl, 25 mM HEPES, 1 mM $MgCl_2$, pH 7.4) and returned to their initial well plate, containing their own conditioned media. After an additional brief period of incubation (15 min to 20 min), the cells were ready for imaging.

### Stimulation

To block neuronal activity, 10 $\mu$M CNQX (Tocris Bioscience, Bristol, UK; Abcam, Cambridge, UK) and 50 $\mu$M AP5 (Tocris Bioscience, Bristol, UK; Abcam, Cambridge, UK) were added to the imaging solution (Tyrode's buffer). Electrical stimulation of the neuronal cultures was performed with field pulses at a frequency of 20 Hz at 20 mA. This stimulation was achieved with 385 Stimulus Isolator (both, World Precision Instruments, Sarasota, FL, USA) and A310 Accupulser Stimulator and with the help of a platinum custom-made plate field stimulator (with 8 mm distance between the plates).

### Imaging

The cells were mounted on a custom-made chamber used for live imaging, containing a pre-warmed imaging solution (Tyrode's solution complemented with the aforementioned drugs). The neurons were then imaged with an inverted Nikon Ti mocroscope, equipped with a cage incubator system (OKOlab, Ottaviano, Italy), Plan Apochromat 60 x 1.4NA oil objective (Nikon Corporation, Chiyoda, Tokyo, Japan), an Andor iXON 897 emCCCD Camera (Oxford Instruments, Andor), with a pixel size of 16 × 16 $\mu$m and Nikon D-LH Halogen 12 V 100 W Light Lamp House. A constant temperature of 37 °C was maintained throughout the imaging procedure. The illumination was 200 ms and the acquisition frequency was 1.7 frames per second (fps). The imaging protocols were conducted as follows: after an initial 10 s baseline, the cells were stimulated for 0.4, 4 or 40 s, followed by a recovery period of 1, 10 or 100 s. In each stimulation, the recovery period was followed by a 2 s test stimulus, followed by 30 s final recovery. The 40 s stimulation set contains an additional longer recovery period of 200 s (not shown in main paper, see SI), resulting in a total of 13 stimulation/recovery conditions. Each condition was conducted on separate coverslips, without repetitions. To image the cumulative of synaptic vesicles, under continual stimulation, the cells were stimulated at 5 Hz, in presence of 1 $\mu$M bafilomycin A1 (purchased from Santa Cruz Biotechnologies).

### Data analysis

The resulting movies were analyzed as follows, using routines generated in Matlab (The Mathworks Inc., Natick MA, USA; version R2022b). The frames were first aligned, to avoid the effects of drift, and they were then summed, to obtain an overall image with a signal-to-noise ratio superior to that of the individual frames. Synapse positions and areas were determined automatically in this image, by a bandpass filtering procedure[66] followed by thresholding, using an empirically-determined threshold. The signal within each of the synapse areas was then determined, for every frame, was corrected for background signal, and was then analyzed by measuring the changes induced by stimulation (as fractional change in intensity, in comparison to the baseline).

## Data availability

Simulation code and experimental data is available on github under github.com/Priesemann-Group/synaptic_power_law_adaptation.

## Code availability

Simulation code and experimental data is available on github under github.com/Priesemann-Group/synaptic_power_law_adaptation.

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

## Acknowledgements

We want to thank the members of the Priesemann Lab, especially Jonas Dehning, for helpful discussions. F.A.M., L.R. S.V.G. and S.O.R. were funded by the German Research Foundation (DFG), SFB1286, Quantitative Synaptology. V.P. received support from the SFB1528, Cognition of Interaction. V.P., S.O.R. and L.R. were supported by the MBExC Excellence Cluster, Deutsche Forschungsgemeinschaft (DFG) under Germany's Excellence Strategy - EXC 2067/1- 390729940. This work was partly supported by the Else Kröner Fresenius Foundation via the Else Kröner Fresenius Center for Optogenetic Therapies.

## Author contributions

F.A.M. Conceptualization, Methodology, Software, Writing - Original Draft; S.V.G. Investigation, Writing - Review & Editing; L.R. Methodology, Writing - Original Draft; S.O.R. Supervision, Writing - Review & Editing; V.P. Conceptualization, Supervision, Writing - Review & Editing.

## Funding

## Competing interests

The authors declare no competing interests.
