## [Transparent Peer Review file · Communications Biology]

Power-law adaptation in the presynaptic vesicle cycle

Corresponding Author: Dr Fabian Mikulasch

This manuscript has been previously reviewed at another journal. This document only contains information relating to versions considered at Communications Biology.

Version 0:

Reviewer comments:

Reviewer #1

(Remarks to the Author)

The manuscript presents a model of short term depression that balances the multiple timescales associated with different stages of vesicle preparation so as to emulate a power-law. The power-law model is compared to a more mechanistic model. A (slightly different) mechanistic model is compared to data in cultured neurons. The functional role of power law short term depression is illustrated using simulation based on naturalistic activity patterns in CA1 cells.

The paper provides a novel point of view that is well anchored in a normative theory (temporal whitening), data (multiple timescales of vesicle processing), and provides a nice complement to the known power law of spike frequency adaptation (Pozzorini et al. Nat Neuro 2013) and power law models of synaptic long term adaptation (Fusi et al. Neuron 2005). The article is well written and the interplay between normative model, detailed model and data re-analysis consolidates the validity of the claims. I quite enjoyed the topic and the treatment. I support its publication in this venue.

There are a number of points I feel are overstatements and a number of technical issues that should be clarified. I hope it is not too inconvenient for the authors if this is presented as a mix of minor and relatively major points:

1. "while synapses typically recover within seconds, (...) synaptic recovery can slow down under strong stimulation" this and the paragraph it is in does not give proper credit to the fact that multi timescale adaptation even at very low time scale is well known in many areas and not necessarily tied to strong stimulation (for instance Salin et al. PNAS 1996). Similarly "models of the function of synaptic adaptation typically simplify the recovery process to a single recovery variable" is hardly justifiable given that even the simplistic Tsodyks Markram model has two recovery variables, and the early models such as Varela et al. (1997) have up to 3. (a similar point about statement on li 41 of p2), and more recent work of Beninger et al. (2024) uses 3 (faster) timescales when they are specifically ignoring the slow processes. I don't think the paper needs to argue that their multiple timescale approach is a novel aspect brought by their effort, the power-law angle is novel enough.
2. Text surrounding the equation on p2 is missing some details. What the i and j refer to is not explained, the terms μ_{ij} and other rates in Fig 1 are not related to r_{ij} . τ_{ij} is not defined in the text.
3. P3, caption "that these very slow processes" which? Fuse+prime? Fuse+prime+docked? Prime+docked?
4. Fig 2E shows a powerlaw model that is not a line on the log-log plot. There must be some kind of mistake.
5. P4 introduces the GLM model, this is a nice effort. I would have liked to have a clearer statement about how related it is to similar phenomenological models with multiple timescales. E.g. Varela et al. J. Neurophys 2007 has one related approach, Rossbroich et al. PLOS CB (2021) has another approach that seem very similar.
6. One critical aspect of the analysis and interpretation of results in Fig 3 is that the question of whether the same exhaustion and pause durations were repeated in a sequence during the experiments. Power law would provide apparent timescales that match the duration of the stimulation when under repeated conditions, something that depends on the effect of the long tail of the power law process. For instance, we would expect that the apparent time scale (Best fit of a single time scale) of the 0.4 data would be 10 times faster than the 40 s. This does not seem to be the case, but it could be because the data did not contain repeated segments with the same duration (it is typical to avoid this by randomly shuffling between conditions). Please clarify.
7. The power law model is missing from Figure 3 which leads to a troubling gap as it means that the power law model is never actually compared to data.
8. "we showed this by fitting the model" specifically what input metric is used to argue this? In a frequentist model this should

be the model MSE compared to the single time scale MSE. But in the Bayesian model, it would be different. Rather the C.I. of the parameters can be used to argue it couldn't be a single timescale. I see the supplementary material uses an overall metric with an acronym, but I could not understand the acronym. It seemed odd that that this model comparison was relegated to the supplementary material as it is one of the main claims of the paper. Please explain the rationale for this conclusion in more details.

9. Fig 3, I could not see if there was any type of cross validation, please specify. Relatedly, whenever relevant, please clarify which plots use cross validation and which do not.

10. Lis 1380164 p7; one should acknowledge that getting to white PSD depends on the input PSD early on and that we don't expect whitening for $1/f$ as the $1/f$ scaling is known for high frequencies, in lower frequency range of the activity typically shows a different regime, revealing a kink in the PSD as in the figure shown. The fact that the paper focuses on frequencies lower than 20 Hz should have been clearer from the beginning: it relates to Short-term plasticity slower than 50 ms or so, since much work on short term plasticity focuses on timescales between 5 ms and 100 ms.

11. "synapse operates" li 158. Synapses don't operate only on this 0.01 Hz to 1 Hz. Range, please modify.

12. Fig.4 panel D is not explained in the caption and does not have a red line.

13. I struggle to understand what authors meant with par 224. The bursting timescale requires short term plasticity of the range of 10s of ms because bursts typically have intervals of 10 to 20 ms. This time scale is ignored here. In fact, again, one should acknowledge that what is treated here could be paired with synapses having STF. This would be important given that CA1 outputs STF (see Friedenberger et al. J. Physiol. 2023).

Reviewer #2

(Remarks to the Author)

This manuscript evaluates the use of power law adaptation along with a multi-stage model of synaptic transmission to determine how sustained synaptic responses are constrained by the many step reactions in the vesicle cycle compared to single stage models. It goes on to show that information flow during sustained activity can be made more efficient by flattening of the output power spectrum of synapses associated with this model.

The model shown does not always accurately describe known kinetic properties of the vesicle release cycle, which also may vary considerably between synapses. For example, endocytosis, the reform step in Fig 1 can be mediated via many mechanisms with different likely rates. These range from kiss and run and ultrafast which may have time courses from tens to hundreds of milliseconds. The loaded pool shown would correspond to the ready releasable pool, but what of reserve pools which may number in the hundreds.

In many or even most cases the model is compared to one with a single timescale. But the latter is clearly too simplistic. We know that even exocytosis is a multi-step process, similarly endocytosis. So it seems like an over simplification to compare a multi step power adaptation model to something so simplistic.

The comparison between the model and experimental data is very limited. This comprises the use of pHluorin attached to a syt1 lumenal Ab and just comparing recovery between to exhaustion protocols. The difference between the single timescale model and the full model is present, but there is no attempt to see what intermediate models might account for the difference. There is also no demonstration of how the model reproduces experimental data under other circumstance – eg even during the exhaustion stimulus itself.

The analysis of temporal whitening is interesting and follows nicely from work cited. However, it is unclear how this can be compared back to what actually happens under physiological conditions. Perhaps if synaptic transmission recorded with the same physiologically derived temporal pattern in cell culture gave similar effects on evoked synaptic release.

The manuscript is extremely sparse in its description of both the model and the associated experimental data making figures quite difficult to understand. Much, if not nearly all of the validation data is reserved for the supplementary information section. But this is only minimally described. For example if S1 the $1/f$ is not labeled other than the thin unlabeled black line, and S7 to S12 are very hard to interpret given the minimal legends.

Some of the wording could be improved. For example, in figure 1A and the associated text, what is meant by further away from the release stage can be inferred as the number of steps in the release cycle model but an explicit statement would help especially with respect to direction in the cycle. So is this slowness a function of endocytosis, reacidification and movement back to the "loaded" pool, or is it affected by other pathways. Even then the quoted speeds (line 78) of up to minutes seems high without including a reserve pool.

Reviewer #3

(Remarks to the Author)

Review of the paper entitled "Power-law adaptation in the presynaptic vesicle cycle"

SUMMARY

In this manuscript, the authors study a published multi-timescale model of synaptic transmission and make 3 claims:

A) The multi-timescale model leads to power-law adaptation

- B) Hippocampal culture data support the multi-timescale model better than the single timescale model
C) Power-law adaptation implies efficient coding through temporal whitening.

The paper addresses a relevant topic. However, I have several concerns regarding the solidity of evidences provided to support claims A and B. The claim C is similar to the one made by Pozzorini [Ref 29], but applied to the synapse dynamics instead of the neuronal dynamics.

MAJOR

1. Solidity of claim A. The authors fitted a GLM on synthetic data generated by the multi-timescale model and found that the effective kernel κ looks consistent with a power-law kernel (Fig. 2E), as one could have expected. However, proving that this kernel κ is effectively power-law is a different business see [1,2]. Here, a proper statistical test is required (since visual inspection of a fitted line on a log-log plot is definitely not sufficient).
2. Solidity of claim B. The authors use (in part) Figure 3E and 3F to show that the hippocampal culture data is well captured by the multiscale model and not by the single timescale model. However, this is far from being obvious when looking carefully at those plots. Indeed, in both plots, the model fits are within the error bars of the experimental data. This is where a proper model comparison is needed. The authors did some efforts in that direction, by providing a Bayesian model comparison in Fig S10. However, there are several points that should be addressed in order to make those results conclusive:
 - a. In Fig S10, there is an overlap between the error bars of the Expected Log Predictive Density (ELPD) of the single timescale model and of the ones of the full model. So is the difference significant? What should be the required difference of the ELPD between the two models such that we can talk about a "very strong" or "decisive" evidence? Did the authors consider natural logarithm or log in base 10 for the ELPD? Please provide the full definition.
 - b. The legend of Fig S10 mentions that the model weights are assigned to 0.64, 0.36 and 0. Those are never defined in the main text nor supplement. Does this mean that the full model has a stronger a prior weight than the single timescale model? But then is this really a fair comparison?
 - c. In our days, a conclusive model comparison relies on several criteria pointing in the same direction whereas here the author rely only on the ELPD. What about other criteria such as BIC or Akaike? Indeed every model comparison method penalises differently for the number of model parameters. This is important here since the full model contains a lot more parameters than the single time scale model.
3. Two timescale model. From Fig S10, it appears that the two timescale model performs almost as well as the full timescale model. This point is also made clear on L195-196. However, the authors argue against the two timescale model for its potential lack of interpretability and for its presupposed inability to capture faster recovery timescales. I find those two arguments rather weak. If we are in the business of performing Bayesian model comparison, we can't disqualify a posteriori a model for its lack of interpretability. I find also tricky to disqualify a model with data that are not used in the model comparison. So if we accept that the two timescale model actually performs well, then the whole framing of the paper on the power-law adaptation is put into question.
4. Deterministic model. In the equation after L281, the authors consider a continuous and deterministic model for the pool sizes P_i . They should discuss under which limit this model is valid. I suspect that when the maximal pool size is small (which is actually the case for the primed pool since $P_{\text{Primed}} = 4$), the stochastic and discrete description is rather different from the continuous and deterministic one since the relative effect of discretisation is large.
5. Likelihood expression. The likelihood defined in the Eq. after L292 needs clarification
 - What is a "condition i "? Does it correspond to the history of spike timing before time t_i ?
 - What is D_i ? Does it correspond to the set of experimentally measured number of released vesicles (as measured by dF/F) for a given spiking history? But then can we really talk about expectation instead of empirical average? How many elements are in the set D_i ? How many conditions in total?
 - The equation assumes the independence between the different conditions (because it is written as a product). The authors should explain why this is the case. If we would consider the stochastic version of the model (instead of the deterministic one), we would not have the independence anymore, right? Please comment.

MINOR

1. Eq after L273. shouldn't $r(t)$ be also in the denominator? The numerator is $\exp(r(t)*a(t))$ whereas the denominator depends on $\exp(a(t))$, without $r(t)$.
2. In Fig S7 and S8 (top), there are no gray lines for the α_{obs} plots.
3. Please number the equations.

REF

[1] Stumpf, Michael P. H., and Mason A. Porter. "Critical Truths About Power Laws." *Science* 335, no. 6069 (February 10, 2012): 665–66. <https://doi.org/10.1126/science.1216142>.

[2] Clauset, Aaron, Cosma Rohilla Shalizi, and M. E. J. Newman. "Power-Law Distributions in Empirical Data." *SIAM Review* 51, no. 4 (November 4, 2009): 661–703. <https://doi.org/10.1137/070710111>.

Version 1:

Reviewer comments:

Reviewer #1

(Remarks to the Author)
I have no further comments.

Reviewer #2

(Remarks to the Author)

Reviewer #3

(Remarks to the Author)

Version 2:

Reviewer comments:

Reviewer #3

(Remarks to the Author)
In the updated version, the authors addressed my concerns. In particular, with the new analysis (which removes the weights w_1 and w_2 and considers instead a hierarchical prior σ_{obs} for the model errors σ_{obs}^1 and σ_{obs}^2), the model comparison (in Fig S11) convincingly favors the GLM power-law over the two time scale model. Actually, I suspect that the tendency of the first analysis to overweight experiment 2 over experiment 1 (which was corrected with those ad-hoc weights) was due to the fact close to $t=0$, the S.E.M is very small, which makes the likelihood extraordinary large if σ_{obs}^2 is small.

Reviewer #1

The manuscript presents a model of short term depression that balances the multiple timescales associated with different stages of vesicle preparation so as to emulate a power-law. The power-law model is compared to a more mechanistic model. A (slightly different) mechanistic model is compared to data in cultured neurons. The functional role of power law short term depression is illustrated using simulation based on naturalistic activity patterns in CA1 cells.

The paper provides a novel point of view that is well anchored in a normative theory (temporal whitening), data (multiple timescales of vesicle processing), and provides a nice complement to the known power law of spike frequency adaptation (Pozzorini et al. Nat Neuro 2013) and power law models of synaptic long term adaptation (Fusi et al. Neuron 2005). The article is well written and the interplay between normative model, detailed model and data re-analysis consolidates the validity of the claims. I quite enjoyed the topic and the treatment. I support its publication in this venue.

Thank you for the summary and constructive comments. We will address them point by point below.

There are a number of points I feel are overstatements and a number of technical issues that should be clarified. I hope it is not too inconvenient for the authors if this is presented as a mix of minor and relatively major points:

1. “while synapses typically recover within seconds, (...) synaptic recovery can slow down under strong stimulation” this and the paragraph it is in does not give proper credit to the fact that multi timescale adaptation even at very low time scale is well known in many areas and not necessarily tied to strong stimulation (for instance Salin et al. PNAS 1996). Similarly “models of the function of synaptic adaptation typically simplify the recovery process to a single recovery variable” is hardly justifiable given that even the simplistic Tsodyks Markram model has two recovery variables, and the early models such as Varela et al. (1997) have up to 3. (a similar point about statement on li 41 of p2), and more recent work of Beninger et al. (2024) uses 3 (faster) timescales when they are specifically ignoring the slow processes. I don't think the paper needs to argue that their multiple timescale approach is a novel aspect brought by their effort, the power-law angle is novel enough.

Thank you for pointing this out. For simplicity, we start with discussing the second point: The first sentence of the paragraph was not formulated very clearly. What we were referring to were models investigating the function, e.g., in the sense of a “normative” theory, which, as far as we know, only employed very simplified recovery dynamics. Therefore, the models by Varela and Beninger, which are important otherwise, are not relevant for this point. Regarding both points: We here are specifically referring to 'recovery' from depression

and deliberately ignored facilitation in the discussion. However, we do agree that it is also slightly imprecise to ignore the fact that these models also employ facilitation, such as the Tsodyks Markram model, which was used for many models of “function”. We cleared up these shortcomings of the old text by making the formulation of the first sentence more precise, and mentioning facilitation as well. The text now reads:

In contrast to these findings, models that investigated what specific functions could be performed by synaptic adaptation typically employed a simplified synaptic model, with one recovery timescale (and, depending on the model, one timescale of facilitation) (Abbott et al. 1997; Goldman et al. 2002; Lavian and Korngreen 2019; Kandaswamy et al. 2010).

Later, we also note that other models (that were not necessarily related to function) previously used more recovery timescales. (Note, that Varela et al employ three recovery timescales in their model, but don’t find evidence for three timescales in their data.):

Many previous studies were set up such that one or two timescales of synaptic recovery could be measured (Varela et al. 1997; Alabi and Tsien 2012; Granseth and Lagnado 2008; Guo et al. 2015), and, consequently, vesicle pool dynamics were typically modeled with only (up to) two recovery timescales (Guo et al. 2015). However, some evidence for more than two relevant timescales of synaptic recovery exists (Rossbroich et al. 2021), which could point to the existence of more relevant timescales of vesicle pool recovery.

2. Text surrounding the equation on p2 is missing some details. What the i and j refer to is not explained, the terms p_{fuse} and other rates in Fig 1 aren’t related to r_{ij} . τ_{ij} is not defined in the text.

Thank you for noticing these shortcomings. We now explain all variables in the text with the equations. The text now reads:

Upon a spike, primed vesicles are released with a certain probability p_{fuse} (in the main results $p_{fuse} = 0.2$). In between spikes, vesicles transition stochastically from pool i to pool j (e.g., docked to primed) with a filling level-dependent transition rate

$$r_{ij} = \frac{c}{\tau_{ij}} \frac{P_i}{P_i^{max}} \left(1 - \frac{P_j}{P_j^{max}} \right), \quad (1)$$

where τ_{ij} is the timescale of vesicle transitions (e.g., τ_{prime}), $c = 4$ ensures that $r_{ij} = 1/\tau_{ij}$ if pools are half full, and P_i is the filling level and P_i^{max} the maximum capacity of pool i .

3. P3, caption ‘that these very slow processes’ which? Fuse+prime? Fuse+prime+docked? Prime+docked?

We meant the ageing and supply processes, which are now referenced directly. These processes were important for the original work in which this model was developed (Jähne et al. 2021), but do not play a major role here, since they unfold over several hours. We added this clarification to the caption.

4. Fig 2E shows a powerlaw model that is not a line on the log-log plot. There must be some kind of mistake.

Thank you for bringing this up. We fitted the kernel with a piecewise-linear function, which we now also mention in the text and the figure legend.

5. P4 introduces the GLM model, this is a nice effort. I would have liked to have a clearer statement about how related it is to similar phenomenological models with multiple timescales. E.g. Varela et al. J. Neurophys 2007 has one related approach, Rossbroich et al. PLOS CB (2021) has another approach that seem very similar.

Thank you for suggesting these models. We were not aware of the model by Rossbroich et al., which is indeed very relevant, as they find a three-timescale recovery process with their model. We now cite this at several relevant locations in the text, in the Introduction and Discussion. The approach to synaptic modeling in both papers is also related, but they use a set of exponential functions instead of bins to parameterize the kernel - we here have the advantage to be able to generate enough data with our model to make this viable. We now mention this in the methods section.

We parameterized the kernel κ through 40 values in log-spaced bins. Maximum-likelihood estimates for parameters $\hat{\theta} = \{\hat{b}, \hat{\kappa}\}$ that describe the simulated data (vesicle releases in response to 1/f spike-trains) were then obtained via gradient ascent on the log-likelihood (SI Fig S17). Similar models of synaptic recovery have been employed before Rossbroich et al. 2021; Varela et al. 1997, which however used a set of exponential functions instead of bins to parameterize the kernel.

6. One critical aspect of the analysis and interpretation of results in Fig 3 is that the question of whether the same exhaustion and pause durations were repeated in a sequence during the experiments. Power law would provide apparent timescales that match the duration of the stimulation when under repeated conditions, something that depends on the effect of the long tail of the power law process. For instance, we would expect that the apparent time scale (Best fit of a single time scale) of the 0.4 data would be 10 times faster than the 40 s. This does not seem to be the case, but it could be because the data did not contain repeated segments with the same duration (it is typical to avoid this by randomly shuffling between conditions). Please clarify.

You are right, the data did not contain repeated segments and hence we did not observe this particular scaling of the recovery timescale. We now clarified this both in figure 3 and the Methods section (Imaging), which reads:

The imaging protocols were conducted as follows: after an initial 10 s baseline, the cells were stimulated for 0.4, 4 or 40 s, followed by a recovery period of 1, 10 or 100 s. In each stimulation, the recovery period was followed by a 2 s test stimulus, followed by 30 s final recovery. The 40 s stimulation set contains an additional longer recovery period of 200 s (not shown in main paper, see SI), resulting in a total of 13 stimulation/recovery conditions. Each condition was conducted on separate coverslips, without repetitions.

7. The power law model is missing from Figure 3 which leads to a troubling gap as it means that the power law model is never actually compared to data.

Thanks for this suggestion. We now also fitted the power law model to the data as a control. For this we introduced a continuous release variant, as for the other models, which we show in SI Fig S5. We show the results of fitting in SI Fig S10. As it turns out, the power law model fits the data from our first experiment reasonably well, and, judged by model comparison, is slightly (but not significantly) better than the two timescales model (SI Fig S11).

8. “we showed this by fitting the model” specifically what input metric is used to argue this? In a frequentist model this should be the model MSE compared to the single time scale MSE. But in the Bayesian model, it would be different. Rather the C.I. of the parameters can be used to argue it couldn't be a single timescale. I see the supplementary material uses an overall metric with an acronym, but I could not understand the acronym. It seemed odd that that this model comparison was relegated to the supplementary material as it is one of the main claims of the paper. Please explain the rationale for this conclusion in more details.

Thank you for suggesting this, it should indeed be much clearer how the model comparison was performed. We here employed leave-one-out (LOO) cross validation (see Vehtari et al. 2017) as implemented by PyMC3. We previously also compared this to the widely applicable information criterion (WAIC) with similar results, but did not include these into the SI. We now include both metrics in the SI, as well as explain the results in more detail in the main paper.

9. Fig 3, I could not see if there was any type of cross validation, please specify. Relatedly, whenever relevant, please clarify which plots use cross validation and which do not.

We used model comparison only in the Bayesian analysis, as mentioned in the question 8. For the GLM in Fig 2 we now also tested against overfitting by measuring the log-likelihood on a test-set (Fig S17). Both tests (for Bayesian model and GLM) are now also mentioned at the relevant points in the text.

10. Lis 1380164 p7; one should acknowledge that getting to white PSD depends on the input PSD early on and that we don't expect whitening for $1/f$ as the $1/f$ scaling is known for high frequencies, in lower frequency range of the activity typically shows a different regime, revealing a kink in the PSD as in the figure shown. The fact that the paper focuses on frequencies lower than 20 Hz should have been clearer from the beginning: it relates to Short-term plasticity slower than 50 ms or so, since much work on short term plasticity focuses on timescales between 5 ms and 100 ms.

This is a good point. We now mention our focus on the longer timescale regime (from several hundreds of milliseconds to several minutes) explicitly when introducing the idea of whitening, where we state:

Natural signals often contain correlations on many timescales, in which case redundancy reduction, i.e., neural adaptation, should also operate on many timescales (Pozzorini et al. 2013). Based on our model, we hypothesized that hippocampal synapses could perform such a redundancy reduction in the longer timescale regime, i.e., from several hundreds of milliseconds to several minutes.

11. "synapse operates" li 158. Synapses don't operate only on this 0.01 Hz to 1 Hz. Range, please modify.

This was indeed not well communicated - we meant to say that here synaptic whitening is most effective. We rectified this phrasing and now refer to the longer timescale regime as introduced above (question 10). The new sentences state:

Note, that perfect $1/f$ scaling over all frequencies is not achievable with a realistic spike train, since it has a finite number of spikes. We therefore generated a signal with $1/f$ scaling in the longer timescale regime where we considered synaptic whitening to be most relevant.

12. Fig.4 panel D is not explained in the caption and does not have a red line.

The figure caption was confusing because we explained both panel C and D together. We now disentangled the descriptions to now read

C Power-law adaptation dynamically adjusts the releases per spike to the input rate. Here we demonstrate this behavior in the strong input regime (~ 4 Hz), which allows for clear visualization. During high activity, releases per spike are selectively reduced, mainly mediated by slow endocytosis. During low activity, they are enhanced when compared to a single timescale model with similar average release rate. The single timescale model closely follows the linear prediction from input spikes (thin red line). D This can be seen more clearly when computing the average number of vesicles released per spike depending on the input spike rate. C,D Rates were estimated by convolving the spike and release trains with kernels (see Methods).

13. I struggle to understand what authors meant with par 224. The bursting timescale requires short term plasticity of the range of 10s of ms because bursts typically have intervals of 10 to 20 ms. This time scale is ignored here. In fact, again, one should acknowledge that what is treated here could be paired with synapses having STF. This would be important given that CA1 outputs STF (see Friedenberger et al. J. Physiol. 2023).

The discussion of bursts of spikes is meant as a hypothetical scenario to illustrate in a simple setting why redundancy reduction has to be adapted to the stimulus. Since we do not necessarily want to refer to actual bursts (with intervals of 10 to 20 ms), but more generally to the concept of clustered spikes that are correlated, we changed the terminology from 'bursts' to 'clusters', and emphasized more that it is a hypothetical scenario. We hope this clears up the confusion of what was meant by the paragraph. It now reads:

Not all spike trains, however, are encoded efficiently by power-law adaptation, and we therefore wondered whether this adaptation is matched to the correlations found in hippocampus. Consider, for example, a (hypothetical) neural code that consists of clusters of spikes separated by intervals of silence with unpredictable length, which only has correlations on the length of the clusters. In this case, synaptic adaptation should only operate on the timescale of clusters, but not longer timescales, as this would reduce responsiveness to the unpredicted (not transmitted) events (Goldman et al. 2002).

Reviewer #2

This manuscript evaluates the use of power law adaptation along with a multi-stage model of synaptic transmission to determine how sustained synaptic responses are constrained by the many step reactions in the vesicle cycle compared to single stage models. It goes on to show that information flow during sustained activity can be made more efficient by flattening of the output power spectrum of synapses associated with this model.

Thank you for the summary and the helpful comments, which we address point by point below.

The model shown does not always accurately describe known kinetic properties of the vesicle release cycle, which also may vary considerably between synapses. For example, endocytosis, the reform step in Fig 1 can be mediated via many mechanisms with different likely rates. These range from kiss and run and ultrafast which may have time courses from tens to hundreds of milliseconds. The loaded pool shown would correspond to the ready releasable pool, but what of reserve pools which may number in the hundreds.

These are good points, and our reasoning here was not clear in the first version of the manuscript. For the endocytosis steps: We took an average kinetic value, measured on populations of synapses, which would be representative for the average behavior of individual synapses. This does not consider variation in the synaptic endocytosis behavior (i.e. one synapse using rapid endocytosis at one point in time, and slower mechanisms at another point), but this type of variation, and its relation to synapse function, is not well known, so it would not be easy to model it. We included this reasoning when introducing the relevant parameter of the model in the main text:

Note, that there might be variation in the rate of endocytosis depending on the mechanism involved (Chanaday et al. 2019), but the extent of this variation under physiological conditions is not well known and we therefor model the average endocytosis rate as measured before (Fähne et al. 2021).

For the reserve pools: The model takes into account both the readily releasable and the recycling pools. Traffic from the reserve to the recycling pool is very slow, taking several days (Truckenbrodt et al. 2018) implying that it does not affect our model.

In many or even most cases the model is compared to one with a single timescale. But the latter is clearly too simplistic. We know that even exocytosis is a multi-step process, similarly endocytosis. So it seems like an over simplification to compare a multi step power adaptation model to something so simplistic.

We agree with the reviewer that a single timescale model is clearly too simplistic to model the recovery dynamics at the synapse, but this is precisely why we compare these models. Many previous models (such as the Tsodyks-Markram model) relied on a single recovery timescale (plus facilitation), and it is important to have such a model as a baseline comparison.

The comparison between the model and experimental data is very limited. This comprises the use of pHLuorin attached to a syt1 luminal Ab and just comparing recovery between to exhaustion protocols. The difference between the single timescale model and the full model is present, but there is no attempt to see what intermediate models might account for the difference. There is also no demonstration of how the model reproduces experimental data under other circumstance – eg even during the exhaustion stimulus itself.

We agree that the single dataset is not sufficient to make conclusive statements, and, as we found before, the data is not sufficient to favor either a two timescales model or the full model via model selection. To address this we recorded another dataset under a different stimulation protocol, where we measured the cumulative release under constant 5 Hz stimulation (Fig 3G-I). We now fit the models to both datasets simultaneously, and find that the multi-timescale model yields the best description of the data, which is also indicated by Bayesian model comparison (SI Fig S11). We discuss this more extensively in the discussion.

The analysis of temporal whitening is interesting and follows nicely from work cited. However, it is unclear how this can be compared back to what actually happens under physiological conditions. Perhaps if synaptic transmission recorded with the same physiologically derived temporal pattern in cell culture gave similar effects on evoked synaptic release.

This is an interesting suggestion, which we considered implementing, but ultimately we could not justify the effort in conducting the experiment with the expected outcome. The problem is that we cannot measure for long enough time, with enough synaptic samples, and with enough temporal precision, to get similar results to those found in the simulation experiments. Since this is not feasible, we did not see any additional information that could be gained by stimulating with physiologically derived temporal patterns over the more controlled experiments we already conducted.

The manuscript is extremely sparse in its description of both the model and the associated experimental data making figures quite difficult to understand. Much, if not nearly all of the validation data is reserved for the supplementary information section. But this is only minimally described. For example if S1 the $1/f$ is not labeled other than the thin unlabeled black line, and S7 to S12 are very hard to interpret given the minimal legends.

We agree that some details and parameters of the model were not well explained. We now took care to introduce every parameter when introducing the model in the main text. We also inserted the missing labels in the SI and improved the captions of the figures to ensure the clarity of the presentation. In the SI Figs S7 to S12 also the axis labels before have not been replaced with easily readable versions, which we now did.

Some of the wording could be improved. For example, in figure 1A and the associated text, what is meant by further away from the release stage can be inferred as the number of steps in the release cycle model but an explicit statement would help especially with respect to direction in the cycle. So is this slowness a function of endocytosis, reacidification and movement back to the “loaded” pool, or is it affected by other pathways. Even then the quoted speeds (line 78) of up to minutes seems high without including a reserve pool.

Thank you for this suggestion, we clarified the mentioned section, which was not well formulated and did not include the whole chain of thought to come to the final conclusion. First, we clarified “distance”. We also clarified the origin of the slowness of some vesicle pools. Since the effective recovery timescale depends on pool size and single vesicle recovery rate, the pool after endocytosis becomes especially slow. The full paragraph now reads:

Intriguingly, the parameters of this model, which have been determined experimentally from cultured hippocampal neurons (Jähne et al. 2021), have a very particular ordering on the cycle (Fig 1A). Transition timescales between pools tend to increase with distance (i.e., number of steps on the vesicle cycle) from release. Vesicle priming can be as fast as 50 ms, while vesicle endocytosis and reformation takes several seconds for a single vesicle. Similarly, the upper limit for the sizes of pools increases with the distance to release. Since the effective recovery timescale of a pool depends on the combination of single vesicle recovery timescales and the pool size limit, these parameters result in very different effective recovery speeds of vesicle pools. Pools close to release recover on fast timescales up to the order of 100 ms, whereas pools further away operate on timescales as slow as 10 s up to minutes (Fig 1B,C).

Reviewer #3

Review of the paper entitled “Power-law adaptation in the presynaptic vesicle cycle”

SUMMARY In this manuscript, the authors study a published multi-timescale model of synaptic transmission and make 3 claims: A) The multi-timescale model leads to power-law adaptation B) Hippocampal culture data support the multi-timescale model better than the single timescale model C) Power-law adaptation implies efficient coding through temporal whitening.

The paper addresses a relevant topic. However, I have several concerns regarding the solidity of evidences provided to support claims A and B. The claim C is similar to the one made by Pozzorini [Ref 29], but applied to the synapse dynamics instead of the neuronal dynamics.

Thank you for the concise summary and the comments. We will address them point by point below.

MAJOR

1. Solidity of claim A. The authors fitted a GLM on synthetic data generated by the multi-timescale model and found that the effective kernel κ looks consistent with a power-law kernel (Fig. 2E), as one could have expected. However, proving that this kernel κ is effectively power-law is a different business see [1,2]. Here, a proper statistical test is required (since visual inspection of a fitted line on a log-log plot is definitely not sufficient).

It is a good idea to make the reasoning that the kernel is best described by a power-law more formal. However, the cited work discusses probability distributions, not general functions, hence we could not employ the statistical tests developed for this purpose. To reinforce our claim we therefore turned to model comparison, and tested several other simple functions as alternative descriptions of the kernel (exponential, stretched exponential, log-normal). Both BIC and AIC identify the truncated power-law as the best model. We have now included

these results in the SI and mention them in the main text. Note, that we now also strengthen this analysis further by testing against overfitting of the kernel with a test set of data.

2. Solidity of claim B. The authors use (in part) Figure 3E and 3F to show that the hippocampal culture data is well captured by the multiscale model and not by the single timescale model. However, this is far from being obvious when looking carefully at those plots. Indeed, in both plots, the model fits are within the error bars of the experimental data. This is where a proper model comparison is needed. The authors did some efforts in that direction, by providing a Bayesian model comparison in Fig S10. However, there are several points that should be addressed in order to make those results conclusive:

a. In Fig S10, there is an overlap between the error bars of the Expected Log Predictive Density (ELPD) of the single timescale model and of the ones of the full model. So is the difference significant? What should be the required difference of the ELPD between the two models such that we can talk about a “very strong” or “decisive” evidence? Did the authors consider natural logarithm or log in base 10 for the ELPD? Please provide the full definition.

We agree that the previous description of model comparison was insufficient to identify which models are favored. To resolve this, we first included another model comparison measure (WAIC) to strengthen the results. Then we provided additional explanations in the SI and main text to clarify the approach and interpretation. The important measure to determine a clear evidence is the standard error of the difference in ELPD, and not the overlap of the standard errors of the ELPD themselves, since the likelihood values are highly correlated between the models. We now indicate this in the caption of the SI figure S10. We furthermore explain the intuitive understanding of the model weights, which give another, more intuitive way of interpreting the results (see below). The relevant part of the caption is

Model comparison of full, single timescale, two timescales, and power-law model
Model comparison computed using leave-one-out (loo) cross validation (top) and the widely applicable information criterion (WAIC, bottom) as implemented by PyMC3 (see Vehtari, A., Gelman, A., & Gabry, J. (2017). Practical Bayesian model evaluation using leave-one-out cross-validation and WAIC. Statistics and computing.). The comparison is based on the expected log pointwise predictive densities (ELPDs), i.e., the expected accuracy on a new dataset, as estimated via WAIC or loo. Plotted are the mean and standard error of the ELPD, and the standard error of the ELPD difference to the best model. In both WAIC and loo, the single timescale, the two timescale, and the power-law model are clearly worse at fitting the data than the multi-timescale model, which is indicated by the standard error of difference in ELPD.

b. The legend of Fig S10 mentions that the model weights are assigned to 0.64, 0.36 and 0. Those are never defined in the main text nor supplement. Does this mean that the full model has a stronger a prior weight than the single timescale model? But then is this really a fair comparison?

Thank you for mentioning this, this was indeed insufficiently explained. Model weights can be thought of as estimated probabilities of a model being the best model on new data (see also McElreath, R. (2016). Statistical Rethinking: A Bayesian Course with Examples in R and Stan. CRC Press). We now define them in the SI in the model comparison figure (Note also that the weights changed with the inclusion of the second experiment):

It is furthermore possible to assign ‘weights’ to the models, which can be thought of as estimated probabilities of a model being the best model on new data. Assigned model weights via both loo and WAIC were 1 for the full model and 0 for the other models, again favoring the full model. For more details on the interpretation of these measures, see McElreath, R. (2016). Statistical Rethinking: A Bayesian Course with Examples in R and Stan. CRC Press

c. In our days, a conclusive model comparison relies on several criteria pointing in the same direction whereas here the authors rely only on the ELPD. What about other criteria such as BIC or Akaike? Indeed every model comparison method penalises differently for the number of model parameters. This is important here since the full model contains a lot more parameters than the single time scale model.

We indeed previously only showed a model comparison based on the ELPD with leave-one-out cross validation. We did also perform a model comparison using the widely applicable information criterion (WAIC) with very similar results, which we now also include in the SI. The BIC or Akaike are not easily applicable to our models in PyMC3, but the WAIC also corrects for the number of parameters in the models in a similar way (See (Vehtari et al. 2017)). We now also mention the usage of both used measures in the main text in the discussion:

In comparison to the full vesicle cycle model, a single timescale recovery model was not able to fit the experimental data as well (Fig 3). This is also the case when comparing the models using Bayesian model comparison, which takes the number of fitted parameters of the models into account (specifically, leave-one-out cross validation and the widely applicable information criterion both clearly favor the full model, see SI Fig S11).

3. Two timescale model. From Fig S10, it appears that the two timescale model performs almost as well as the full timescale model. This point is also made clear on L195-196. However, the authors argue against the two timescale model for its potential lack of interpretability and for its presupposed inability to capture faster recovery timescales. I find those two arguments rather weak. If we are in the business of performing Bayesian model comparison, we can't disqualify a posteriori a model for its lack of interpretability. I find also tricky to disqualify a model with data that are not used in the model comparison. So if we accept that the two timescale model actually performs well, then the whole framing of the paper on the power-law adaptation is put into question.

These are valid points, and we addressed this by including data from a second experiment to fit the models (Fig 3G-I). This new data shows that the two timescales model does not fit the data as well as the full model, which we showed using Bayesian model comparison. While it can capture different recovery timescales depending on the stimulation, it does not capture the full diversity found in experiment. We now discuss this in the discussion

Bayesian model comparison also favors the full model over a model with two timescales of vesicle recovery (SI Fig S9,S10). While the two timescales model can succeed in modeling a change of recovery timescale depending on the stimulation, it failed to describe the data from both experiments simultaneously as well as the full model (SI Fig S9). Mostly, it seems that the different experimental conditions require more diversity in recovery timescales than the two timescales model can provide. Still, while our experiments test recovery on medium and longer timescales, they do not provide data on different synaptic recovery on faster timescales, which are predicted by our model. Although these shorter timescales have been found in other experiments Hanse and Gustafsson 2001; Varela et al. 1997; Rossbroich et al. 2021, a full experimental validation of the model would require a different experimental setup that can measure very short, medium, and long timescales of synaptic recovery simultaneously.

4. Deterministic model. In the equation after L281, the authors consider a continuous and deterministic model for the pool sizes P_i . They should discuss under which limit this model is valid. I suspect that when the maximal pool size is small (which is actually the case for the primed pool since $P_{\text{Primed}} = 4$), the stochastic and discrete description is rather different from the continuous and deterministic one since the relative effect of discretisation is large.

It is true that for small vesicle number the behavior on single realizations is very different from the continuous model. However, averaging over many realizations, we expect the behavior to be mostly the same, as here the average pool filling levels should converge to the continuous variant. We previously verified this with figure S5 in the SI, which demonstrated that depression behaves essentially the same. Note, that we found a small error in this previous comparison, in how the vesicles were counted in the bins. Correcting this error shows that the two model align much better than shown before. In addition, we now extended this figure to verify that under the particular experimental protocol we find agreeing model predictions. This new figure shows that in the particular experimental setup and recording paradigm, both models give virtually identical results.

5. Likelihood expression. The likelihood defined in the Eq. after L292 needs clarification

- What is a “condition i ”? Does it correspond to the history of spike timing before time t_i ?
- What is D_i ? Does it correspond to the set of experimentally measured number of released vesicles (as measured by dF/F) for a given spiking history? But then can we really talk about expectation instead of empirical average? How many elements are in the set D_i ? How many conditions in total?
- The equation assumes the independence between the different conditions (because it is written as a product). The authors should explain why this is the case. If we would consider the stochastic version of the model (instead of the deterministic one), we would not have the independence anymore, right? Please comment.

Thank you for pointing out the imprecise description.

- One condition is a particular combination of exhaustion and pause time, which we now clarified.

- D_i is the recorded increases in fluorescence ($\Delta F/F$ change) for all synapses in one condition, which is now clarified. The faulty notation was changed from expectation to empirical average. We also explicitly noted all conditions down. We furthermore corrected a notation error where we noted the observed data as the likelihood mean instead of the sampled simulation data.

- The independence is given because all conditions were conducted on independent samples, which we now also make more clear in the Imaging subsection. Therefore, in both the deterministic case and the stochastic case errors would be expected to be independent. The reasoning has been included when introducing the Likelihood.

The full description (including the changes made to also model the second experiment) is now

Formally, we computed the log posterior probability of the parameters θ given the experimental data of both experiments $D^{(1)} = \{D_i : i \in \text{conditions}\}$, $D^{(2)} = \{D_t^{(2)} : t \in \{0..T\}\}$:

$$\log p(\theta|D^{(1)}, D^{(2)}) \propto w^{(1)} \log p(D^{(1)}|\theta) + w^{(2)} \log p(D^{(2)}|\theta) + \log p(\theta), \quad (2)$$

where $w^{(1)}$ and $w^{(2)}$ are used to re-weight data as explained below.

In experiment 1, the data $D_i^{(1)}$ are the recorded increases in fluorescence ($\Delta F/F$ change) for all synapses in one condition i , and the conditions are all tested combinations of exhaustion and pause time, i.e., conditions = $\{(T_{exh}, T_{pause}) : T_{exh} \in \{0.4 \text{ s}, 4 \text{ s}, 40 \text{ s}\}, T_{pause} \in \{1 \text{ s}, 10 \text{ s}, 40 \text{ s}, 100 \text{ s}\}\}$. Since each experimental condition was tested on independent synaptic samples (see Imaging), the likelihood was defined as a product of Gaussian likelihoods for each condition

$$p(D^{(1)}|\theta) = \prod_{i \in \text{cond.}} \mathcal{N}(\bar{D}_i^{(1)}; \mu = \alpha_{obs}^{(1)} O_i(\theta_{obs}), \sigma = \sigma_{obs}^{(1)} + SEM(D_i^{(1)})), \quad (3)$$

where $\bar{D}_i^{(1)}$ is the empirical mean over recorded synapses, $SEM(D_i^{(1)})$ is the error of the mean, and $O_i(\theta_{obs})$ is the number of releases measured in the simulation under stimulation condition i and parameters θ_{obs} .

In experiment 2, the data $D_t^{(2)}$ are the fluorescence traces measured in 1231 synapses over time. Again, assuming independence of experimental noise, the likelihood was defined as a product of Gaussian likelihoods for each time point

$$p(D^{(2)}|\theta) = \prod_{t \in \{0..T\}} \mathcal{N}(\bar{D}_t^{(2)}; \mu = \alpha_{obs}^{(2)} O_t(\theta_{obs}), \sigma = \sigma_{obs}^{(2)} + SEM(D_t^{(2)})), \quad (4)$$

with most variables defined similar to experiment 1.

MINOR

1. Eq after L273. shouldn't $r(t)$ be also in the denominator? The numerator is $\exp(r(t)*a(t))$ whereas the denominator depends on $\exp(a(t))$, without $r(t)$.

Thank you for examining this in detail, we checked this, but the equation is in fact correct. Another way to write this model is as $p(r|a)=1/Z * \exp(r*a)$, where Z is the sum over all r to normalize the sum over all $p(r|a)$ to 1. If r would be in the denominator, $p(0|a)=1/2$ always.

2. In Fig S7 and S8 (top), there are no gray lines for the alpha_obs plots.

We defined an (improper) flat prior on alpha_obs, hence there is no prior to be shown. We explain this now in the caption.

3. Please number the equations.

The equations are now numbered.

REF

[1] Stumpf, Michael P. H., and Mason A. Porter. "Critical Truths About Power Laws." *Science* 335, no. 6069 (February 10, 2012): 665–66. <https://doi.org/10.1126/science.1216142>.

[2] Clauset, Aaron, Cosma Rohilla Shalizi, and M. E. J. Newman. "Power-Law Distributions in Empirical Data." *SIAM Review* 51, no. 4 (November 4, 2009): 661–703. <https://doi.org/10.1137/070710111>.

References

- Abbott, Larry F et al. (1997). "Synaptic depression and cortical gain control." In: *Science* 275.5297, pp. 221–224.
- Alabi, AbdulRasheed A and Richard W Tsien (2012). "Synaptic vesicle pools and dynamics." In: *Cold Spring Harbor perspectives in biology* 4.8, a013680.
- Chanaday, Natali L et al. (2019). "The synaptic vesicle cycle revisited: new insights into the modes and mechanisms." In: *Journal of Neuroscience* 39.42, pp. 8209–8216.
- Goldman, Mark S, Pedro Maldonado, and LF Abbott (2002). "Redundancy reduction and sustained firing with stochastic depressing synapses." In: *Journal of Neuroscience* 22.2, pp. 584–591.
- Granseth, Björn and Leon Lagnado (2008). "The role of endocytosis in regulating the strength of hippocampal synapses." In: *The Journal of physiology* 586.24, pp. 5969–5982.
- Guo, Jun et al. (2015). "A three-pool model dissecting readily releasable pool replenishment at the calyx of held." In: *Scientific reports* 5.1, p. 9517.
- Hanse, Eric and Bengt Gustafsson (2001). "Vesicle release probability and pre-primed pool at glutamatergic synapses in area CA1 of the rat neonatal hippocampus." In: *The Journal of Physiology* 531.2, pp. 481–493.

-
- Jähne, Sebastian et al. (2021). “Presynaptic activity and protein turnover are correlated at the single-synapse level.” In: *Cell Reports* 34.11, p. 108841.
- Kandaswamy, Umasankar et al. (2010). “The role of presynaptic dynamics in processing of natural spike trains in hippocampal synapses.” In: *Journal of Neuroscience* 30.47, pp. 15904–15914.
- Lavian, Hagar and Alon Korngreen (2019). “Short-term depression shapes information transmission in a constitutively active GABAergic synapse.” In: *Scientific Reports* 9.1, p. 18092.
- Pozzorini, Christian et al. (2013). “Temporal whitening by power-law adaptation in neocortical neurons.” In: *Nature neuroscience* 16.7, pp. 942–948.
- Rossbroich, Julian et al. (2021). “Linear-nonlinear cascades capture synaptic dynamics.” In: *PLoS computational biology* 17.3, e1008013.
- Truckenbrodt, Sven et al. (2018). “Newly produced synaptic vesicle proteins are preferentially used in synaptic transmission.” In: *The EMBO journal* 37.15, e98044.
- Varela, Juan A et al. (1997). “A quantitative description of short-term plasticity at excitatory synapses in layer 2/3 of rat primary visual cortex.” In: *Journal of Neuroscience* 17.20, pp. 7926–7940.
- Vehtari, Aki, Andrew Gelman, and Jonah Gabry (2017). “Practical Bayesian model evaluation using leave-one-out cross-validation and WAIC.” In: *Statistics and computing* 27, pp. 1413–1432.

Reviewer #2 (Remarks to the Author):

This is a resubmitted manuscript which evaluates the use of power law adaptation along with a multi-stage model of synaptic transmission to determine how sustained synaptic responses are constrained by the many step reactions in the vesicle cycle. I believe this is a useful set of findings.

There were extensive initial comments from reviewers in the prior round of review that the authors have responded to carefully.

I am concerned still about two points.

1) That the model conflates many likely mechanisms of synaptic vesicle endo and exocytosis. The authors argue in their response that “For the endocytosis steps: We took an average kinetic value, measured on populations of synapses, which would be representative for the average behavior of individual synapses. This does not consider variation in the synaptic endocytosis behavior (i.e. one synapse using rapid endocytosis at one point in time, and slower mechanisms at another point), but this type of variation, and its relation to synapse function, is not well known, so it would not be easy to model it”

However, it is very likely that different frequencies and durations of stimulation generate different modes of exo/endocytosis. So, taking the average response may conflate critical alterations in recovery times and principal mechanisms. For example, a burst of high frequency stimulation might trigger bulk endocytosis. I am concerned that this can be averaged with slower stimulation changes. I appreciate the point that the stimulation is really clusters of spikes of variable intervals, but it would be surprising if it made no difference how responses differed between and across clusters. I was not so much concerned of each synapse having a different mechanism of recycling, but this being a variable at each synapse. On this point the authors clarification to reviewer one on the range of stimulus timing (not very high frequencies) helped on this point but an acknowledgment of this limitation in the text and how it might effect modes of recycling would be helpful.

Thank you for this comment, a more thorough discussion of these assumptions is indeed warranted to point out the limitations of the model. We now discuss this more thoroughly at two points in the manuscript. First, in the results section when the model is introduced, where we extended the previous discussion of the topic:

Note, that there might be variation in the rate of endocytosis depending on the mechanism involved or the precise stimulation, for example the presence of bulk endocytosis under strong stimulation (Chanaday et al. 2019). However, the extent of this variation under physiological conditions is not well known and we therefore model the average endocytosis rate as measured before (Jähne et al. 2021). The validity of this approach relies on the assumption that under the moderate stimulation we will use in our simulations, and due to the generally relatively slow rate of endocytosis, these effects play a limited role in determining synaptic coding function.

And second, in the discussion section, where we mention this limitation along the discussion of the possible role of short-term facilitation in determining synaptic coding function:

At the synapse, such a robust code (or other types of codes) could for example be achieved through short-term facilitation (Mahajan and Nadkarni 2020), or different, activity dependent endocytosis mechanisms (Chanaday et al. 2019), which we did not model here. With a better understanding of these processes, it will be important to consider how they can interact with multi-stage vesicle dynamics to shape synaptic function.

2) The authors also argue that they cannot generate real data with the timing of used to generate conclusions relating to whitening. I am not entirely convinced this is not possible if approaches beyond pHluorin were considered.

You are right that electrophysiological recordings would, indeed, provide a read-out that is more flexible, and can address widely different timing conditions. However, first, such recordings fail to provide direct information on the exo- and endocytosis responses. Thus, they are not immediately useful for the analysis as we conducted it here, modeling the presynaptic response. Second, well funded claims on whitening would require several hours of high temporal resolution recordings of synaptic release, at individual synapses, and we are not aware of any recording technique that would allow for this. As much as we would like to directly test the idea of temporal whitening experimentally, currently this seems to be beyond of what we can achieve with our experimental model, hippocampal cultured neurons.

Reviewer #3 (Remarks to the Author):

The authors partially addressed my concerns. On the positive side, I appreciate that they included additional model comparison metric such as WAIC as well as considered additional piece of data (see Fig. 3G-I). However, there are still critical problems with the current manuscript, especially the issue 1 below.

1. My main concern is that the present manuscript is framed around power-law and I am still not convinced that the power-law adaptation model is better than the two-time scale model. This for the following reasons:

a. In response to my major comment 1, the authors performed a model comparison with the AIC and BIC metric. In particular, they fitted GLMs with various kernels (power-law, log-normal, exponential, stretched exponential) to synthetic data obtained from the full model. However, they didn't use a double-exponential kernel. So it remains unclear whether a double-exponential kernel (that would capture a two time-constant model) would be better or worse than the power-law kernel.

Thank you for this suggestion, we now also tested the double exponential function. As before, we found that both AIC and BIC indicate that the power-law function provides the best fit of the kernel. We have now added these new results to the Supplementary Information (Fig S17). For transparency, note that we also found a small inconsistency in our previous approach, that we corrected: While the fitting and plotting of the kernel was done in log-space, we computed the log-likelihood in normal space. To be consistent, we now also compute the log-likelihood in log-space. This does change the absolute values of the AIC and BIC, but does not change the results we found.

b. Regarding the fit to experimental data, the authors additionally included in this revision the evolution of the fluorescence as a function of time (Fig 3G-I). Even-though this additional piece of data provides good evidence against the single time-scale model (Fig 3I), it is not clear at all whether it provides evidence for the power-law model (Fig. 3H) over the two-time scale model (Fig S9, bottom-right). So the statement on L219 ("While the two timescales model can succeed in modeling a change of recovery timescale depending on the stimulation, it failed to describe the data from both experiments simultaneously as well as the full model (SI Fig S9).") is unsubstantiated and misleading.

Before we answer this question, please note that we changed our analysis (see your question 2). With this change the difference between the models became more pronounced. While this can be difficult to judge from the fits alone, it is easily visible from the model comparison (Fig S11). We now reference this figure at the cited section of the discussion, to justify the statement. We furthermore tried to be more specific as to why we think the two time-scale model does not fit as well. The updated sentence in the discussion now reads:

While the two timescales model can succeed in modeling a change of recovery timescale depending on the stimulation, it failed to describe the data from both experiments simultaneously as well as the full model (SI Fig S9,S11). It seems that there are different requirements between the multi timescale depression (Fig 3E) and the need to release enough vesicles to saturate the fluorescence signal (Fig 3H), that lead to contradictions within the single and two-timescale models.

c. Fig S11, the model comparison on Fig S11 shows almost no difference between the power-law and the two-timescale model, both for the ELPD_LOO as well as for the ELPD_WAIC.

This is a real issue since the whole manuscript is frame under the lenses of power-law, starting from the title itself. So either the authors make a convincing case for power-law (which is currently not the case) or they reframe completely the story.

You are right in pointing out this shortcoming of the GLM model. However, as we noted before and as we will discuss in question 2, our results changed with the new analysis, and the power-law GLM is now on par with the full model and clearly better than the two-timescale model.

Why does the GLM fit the data better in our new analysis? There are three main changes that allow for this: a) The second dataset is weighted more: this constrains model parameters more strongly; b) we allow the model fluorescence signal to saturate under enough released vesicles: this means the synapse no longer has to depress to failure to model the second dataset; c) we coupled the model errors of the two experiments: this means that if one dataset is modeled well, the other dataset cannot be ignored in the fit. Together these changes constrained the GLM much more, which allowed for a better fit.

This solves the main criticism, but we found it nevertheless useful to point out a shortcoming of the GLM in modeling the deep depression of the synapse, which is inherent in its linear form, and which was previously the reason why the full model and the GLM model did not match in performance. In the discussion we thus write:

Under moderate synaptic stimulation this recovery can be described by a generalized linear model with an effective power-law depression kernel up to about 120 s (Fig 2E). A shortcoming of this linear description is that it does not capture the deep depression of the synapse under very strong stimulation, which can occur in the vesicle pool model (SI Fig S5). Nevertheless, for the functionally more relevant moderate stimulation regime, the power-law adaptation model yields a simplified description of the cascading dynamics of vesicle pool recovery.

2. In Eq. 6, the authors reweigh the log-likelihood with weights w^1 and w^2 . They justify this procedure by the fact that there are different number of data points in D^1 and D^2 . I don't understand the logic here. Log-likelihoods automatically scale with the number of data points. For independent observations $D = \{y_i\}_{i=1:N}$, we have $\log p(D|\theta) = \sum_{i=1}^n \log p(y_i|\theta)$. As a consequence for any split of the data $D = D^1 \cup D^2 = \{y_i\}_{i=1:n} \cup \{y_i\}_{i=n+1:N}$, we have $\log p(D^1, D^2|\theta) = \log p(D^1|\theta) + \log p(D^2|\theta)$ such that there is no need to reweigh.

You are right that the reweighing is a non-standard approach in Bayesian Inference, which is ad-hoc and not well justified. To reiterate, we introduced this as a way to combat the tendency of the sampling algorithm to ignore the first dataset while finding the posterior—essentially, since the contribution of this dataset to the overall likelihood would be very small, the algorithm would not find the global minimum and thus a good fit.

To pursue a more justified Bayesian approach, we decided to remove the weighting and instead make another reasonable assumption in the model: Since both experiments model the same synapses, we assumed that the model error (which is added to the empirical measurement error per experiment) is approximately equal for both experiments. This can be formulated as a simple hierarchical prior for the model error. This prevents the sampling algorithm to assign an unreasonably high model error to the first experiment, and thus effectively ignore it. We updated the methods:

Using MCMC, we then estimated the model parameters $\theta = \{\alpha_{obs}^{(1)}, \alpha_{obs}^{(2)}, \sigma_{obs}^{(1)}, \sigma_{obs}^{(2)}, \sigma_{obs}\} \cup \theta_{obs}$, where the α_{obs} are factors that scale the number of released vesicles (model) to the increase in dF/F (experiment), σ_{obs} are the model errors (see below), and $\theta_{obs} = \{p_{fuse}^{(1)}, p_{fuse}^{(2)}, r_{ij}, F_i^{max} : i, j \in Pools\}$ are the vesicle cycle parameters. Here, the α_{obs} and p_{fuse} were used to model experiment 1 or 2 (Fig 3), respectively. Note, that we employ a hierarchical prior σ_{obs} for the model errors $\sigma_{obs}^{(1)}$ and $\sigma_{obs}^{(2)}$ (for details see SI table 1), since we expect both experiments to be fitted similarly well by the models, and as a way to prevent any experiment from dominating the likelihood with a very small inferred error. Also note, that the width of the likelihood will also depend on the empirical measurement error, as explained below.

Another problem of our previous approach was that we assumed that the plateauing fluorescence signal in the second experiment results from a failure of the synapse to release new vesicles. However, this would constitute an extremely strong depression which is unlikely under this short 5 Hz stimulation. Instead, we found that with the correct data weighting the full and the GLM model actually recovered the likely "correct" solution, which is that the fluorescence saturates when all vesicles have been released once. Overall, this allows for a much better and more plausible fit of the two experiments.

We updated all results accordingly. We also updated the methods, where previously we mentioned that modeling the saturation does not make a difference in the results, as the inferred release rates were too small. As mentioned, this is now no longer the case, and we now simply write:

$O_t(\theta_{obs}) := n_t$ is the estimated number of vesicles that have been released once (i.e., are fluorescent). Under the assumption that previously released and unreleased vesicles are well mixed, this number can be computed as $n_{t+1} = n_t + r_t f_t$, where $f_t = 1 - n_t/N$ is the fraction of unreleased vesicles, N is the total number of vesicles, and r_t is the number of vesicles emitted at t . This ensures that the model does not predict more fluorescent vesicles than there are total vesicles.

One notable difference to the results before (next to the differences discussed in the previous answers) is that also the single and two timescale models now have somewhat different posterior distributions of model parameters. The reason is that now the second experiment is weighted more strongly. For both this means that the second experiment fit is improved at the cost of the fit of the first experiment. Additionally, some fitted parameters become extremely implausible, and we changed the relevant section in the results part, which now reads:

Furthermore, the cumulative response of the synapse to constant stimulation starts to plateau after 1 min of stimulation, which is well captured by the full model (Fig 3H). Also the single timescale model can fit this data reasonably well (Fig 3I), but it requires extremely low release probabilities ($p_{fuse} \approx 0.01$) and slow recovery rates to achieve this (SI Fig S8).

Another small difference we have to mention occurs in the model with unconstrained transition rates, which we show as a model validation in the SI (Fig S14). Here, with the new approach, the model surprisingly recovers all transition rates from the full model, which

are very close to the experimentally motivated parameters. This provides an unexpected but very nice further validation of the model. We note this in the discussion:

Still, while our experiments test recovery on medium and longer timescales, they do provide little data on synaptic recovery on faster timescales, which are predicted by our model. Although these shorter timescales have been found in other experiments Hanse and Gustafsson 2001; Varela et al. 1997; Rossbroich et al. 2021, a full experimental validation of the model would require a different experimental setup that can measure very short, medium, and long timescales of synaptic recovery simultaneously. Surprisingly, however, when fitting the full model without precise priors on the recovery timescales, the model recovers all timescales of recovery from the data alone (SI Fig S14), which constitutes an indirect validation of also the faster recovery dynamics.

Minor

1. L306 It is stated that the kernel kappa is parametrised through 40 values in log-spaced bins. Please provide the explicit bin sizes and positions.

We added the explicit bins to the methods:

We parameterized the kernel κ through 40 values in log-spaced bins (with boundaries $\{e^q : q \in \{\log(a_{min}) + j/39(\log(a_{max}) - \log(a_{min})) : j \in [0..39]\}\}$), where $a_{min} = 0.001$ s, $a_{max} = 500$ s).

2. The horizontal axis of many plots are not scaled adequately. In particular the plots showing the distribution of sigma_obs in the supplementary figures (S7, S8, S9, S10, S12, S14)

Thank you for noticing this. We now display the posteriors more clearly.

Note: all the line numbers are taken from the new manuscript (where the corrections are not highlighted in blue).

References

- Chanaday, Natali L et al. (2019). "The synaptic vesicle cycle revisited: new insights into the modes and mechanisms." In: *Journal of Neuroscience* 39.42, pp. 8209–8216.
- Hanse, Eric and Bengt Gustafsson (2001). "Vesicle release probability and pre-primed pool at glutamatergic synapses in area CA1 of the rat neonatal hippocampus." In: *The Journal of Physiology* 531.2, pp. 481–493.
- Jähne, Sebastian et al. (2021). "Presynaptic activity and protein turnover are correlated at the single-synapse level." In: *Cell Reports* 34.11, p. 108841.
- Mahajan, Gaurang and Suhita Nadkarni (2020). "Local design principles at hippocampal synapses revealed by an energy-information trade-off." In: *Eneuro* 7.5.
- Rossbroich, Julian et al. (2021). "Linear-nonlinear cascades capture synaptic dynamics." In: *PLoS computational biology* 17.3, e1008013.
- Varela, Juan A et al. (1997). "A quantitative description of short-term plasticity at excitatory synapses in layer 2/3 of rat primary visual cortex." In: *Journal of Neuroscience* 17.20, pp. 7926–7940.